# The eukaryotic translation initiation factor eIF4E harnesses hyaluronan production to drive its malignant activity

**Hiba Ahmad Zahreddine[1], Biljana Culjkovic-Kraljacic[1], Audrey Emond[2], Filippa Pettersson[2], Ronald Midura[3,4], Mark Lauer[3,4], Sonia Del Rincon[2], Valbona Cali[3,4], Sarit Assouline[2], Wilson H Miller[2], Vincent Hascall[3,4], Katherine LB Borden[1]\***

[1]Department of Pathology and Cell Biology, Institute for Research in Immunology and Cancer, Université de Montréal, Québec, Canada; [2]Segal Cancer Centre, Lady Davis Institute, Jewish General Hospital, McGill University, Québec, Canada; [3]Orthopaedic Research Center, The Cleveland Clinic Foundation, Cleveland, United States; [4]Department of Biomedical Engineering, Lerner Research Institute, The Cleveland Clinic Foundation, Cleveland, United States

**Abstract** The microenvironment provides a functional substratum supporting tumour growth. Hyaluronan (HA) is a major component of this structure. While the role of HA in malignancy is well-defined, the mechanisms driving its biosynthesis in cancer are poorly understood. We show that the eukaryotic translation initiation factor eIF4E, an oncoprotein, drives HA biosynthesis. eIF4E stimulates production of enzymes that synthesize the building blocks of HA, UDP-Glucuronic acid and UDP-N-Acetyl-Glucosamine, as well as hyaluronic acid synthase which forms the disaccharide chain. Strikingly, eIF4E inhibition alone repressed HA levels as effectively as directly targeting HA with hyaluronidase. Unusually, HA was retained on the surface of high-eIF4E cells, rather than being extruded into the extracellular space. Surface-associated HA was required for eIF4E's oncogenic activities suggesting that eIF4E potentiates an oncogenic HA program. These studies provide unique insights into the mechanisms driving HA production and demonstrate that an oncoprotein can co-opt HA biosynthesis to drive malignancy.
DOI: https://doi.org/10.7554/eLife.29830.001

**\*For correspondence:**
katherine.borden@umontreal.ca

**Competing interests:** The authors declare that no competing interests exist.

## Introduction

The tumor microenvironment plays important roles in cancer, providing a niche for the preferential survival and proliferation of tumor cells. A major component of this structure is the glycosaminoglycan hyaluronan (HA). HA is composed of repeating disaccharides units of UDP-Glucuronic Acid and UDP-N-Acetyl Glucosamine. HA is synthesized by hyaluronic acid synthases (HAS), which are single transmembrane proteins localized to the inner-face of the plasma membrane (*Kultti et al., 2006*; *Lenart et al., 2017*). HA chain length can be associated with differential functions including regulation of various cellular processes such as embryonic development, tissue homeostasis, wound healing and inflammation and when dysregulated can promote EMT, tumor growth, and invasion (*Goncharova et al., 2012*; *Toole, 2004*). Shorter chains are primarily synthesized by HAS3 (*Itano et al., 1999*; *Spicer and McDonald, 1998*). These shorter forms of HA are often associated with malignant phenotypes (*Slevin et al., 2007*; *Slevin et al., 2002*; *West et al., 1985*). Artificial overexpression of HAS enzymes causes increased tumor growth in mouse xenograft models of prostate, breast and colon carcinomas while its knockdown reverses this phenotype (*Kultti et al., 2006*; *Koistinen et al., 2015*). Further, HA is cleaved by hyaluronidases which have been suggested to act

as tumor suppressors; whereby increased expression inhibits tumor growth in colon and breast xenografts (*Bertrand et al., 2005*; *Junker et al., 2003*; *Sá et al., 2015*). Increased HA levels are correlated with formation of less dense matrices to facilitate invasion and promote angiogenesis. Elevated levels of HA in the stroma around tumors is associated with poor outcome (*Anttila et al., 2000*; *Auvinen et al., 2000*). In addition to surrounding tumors in some cases, HA can coat the surface of tumor cells with HA-based protrusions radiating from the cell surface (*Ropponen et al., 1998*; *Setälä et al., 1999*). Indeed, overexpression of HAS3 alone is sufficient to induce the formation of an HA coat with microvillus-like protrusions on the cell surface. The major HA receptor CD44 is found to co-localize with these surface HA coats but is not required for HA coat formation. The number of cancer cell types with cell-surface HA is not yet known, and the extent to which cell-associated HA also plays physiological roles in cancer is an important open question in the field.

Despite the wealth of knowledge relating HA to malignancy, there is virtually no information regarding how HA levels become elevated in cancer and further, there is no understanding of what physiological conditions drive production of cell-surface-associated HA. Indeed, the levels of mRNAs encoding the enzymes in the biosynthetic pathways are often poor predictors of HA production (*Nykopp et al., 2010*). For instance in endometrial and ovarian carcinomas, mRNA levels of HAS enzymes do not predict elevated HA levels in these specimens (*Nykopp et al., 2010*; *Nykopp et al., 2009*). These data suggest that this pathway is not always under direct transcriptional control and begs the question how else is HA production regulated. In this study, we demonstrate that the production of HA and its related downstream effectors are coordinately controlled post-transcriptionally by the eukaryotic translation initiation factor eIF4E. eIF4E is highly expressed in many cancers and this correlates with increased invasion, metastases and poor prognosis (*Assouline et al., 2015*; *Culjkovic-Kraljacic and Borden, 2013*; *Gao et al., 2016*; *Pettersson et al., 2015*; *Xu et al., 2016*). In early-stage clinical trials, eIF4E targeting with ribavirin led to objective responses including remissions in some acute myeloid leukemia (AML) patients (*Assouline et al., 2009*; *Assouline et al., 2015*). In mouse models, eIF4E overexpression is sufficient to drive tumor formation (*Kentsis et al., 2004*; *Lazaris-Karatzas et al., 1990*; *Lazaris-Karatzas and Sonenberg, 1992*). At the biochemical level, eIF4E modulates expression of selected transcripts through its roles in nuclear mRNA export and translation. Both these functions contribute to its oncogenic potential (*Culjkovic-Kraljacic and Borden, 2013*; *Carroll and Borden, 2013*; *Osborne and Borden, 2015*). eIF4E-target transcripts are the downstream effectors of its physiological effects. Here, we identified the enzymes encoding the HA biosynthetic pathway, HAS3, CD44 and associated factors as eIF4E target transcripts. We demonstrated that this pathway was required for eIF4E to mediate its oncogenic activities.

## Results and discussion

We set out to identify mRNA target transcripts which could encode proteins that were downstream effectors of the oncogenic activities of eIF4E. To identify these mRNAs, we took advantage of a fundamental difference in the RNA-binding properties of eIF4E between cellular compartments. In the cytoplasm, eIF4E binds all capped-RNAs regardless of whether it increases their translation efficiency (*Rousseau et al., 1996*; *Culjkovic-Kraljacic et al., 2016*; *Graff and Zimmer, 2003*). However, in the nucleus, eIF4E binds transcripts that are functional export targets (*Culjkovic et al., 2005*; *Culjkovic et al., 2006*). These export target mRNAs typically contain a structurally-defined 50-nucleotide element known as an eIF4E sensitivity element (4ESE) in their 3' UTR and are $m^7G$ capped (*Culjkovic et al., 2006*, *2007*). This element is directly bound by the leucine-rich pentatricopeptide repeat (LRPPRC) protein which also directly binds the dorsal surface of eIF4E. The eIF4E-4ESE RNA-LRPPRC complex in turn binds the CRM1 nuclear pore receptor which transits it to the cytoplasm (*Topisirovic et al., 2009*; *Volpon et al., 2016*). To date, about 3500 RNAs have been shown to immunoprecipitate with eIF4E from the nuclear fraction of lymphoma cells suggesting that this is a broadly used mRNA export pathway (*Culjkovic-Kraljacic et al., 2016*). Given these considerations, we reasoned that identification of eIF4E-bound transcripts in the nucleus would provide a straightforward strategy for the discovery of downstream factors that execute its biological effects. These included enzymes involved in HA biosynthesis (see below).

We identified eIF4E-bound RNAs using an RNA immunoprecipitation (RIP) strategy. Nuclear lysates from osteosarcoma U2Os cells were immunoprecipitated with anti-eIF4E antibodies, and

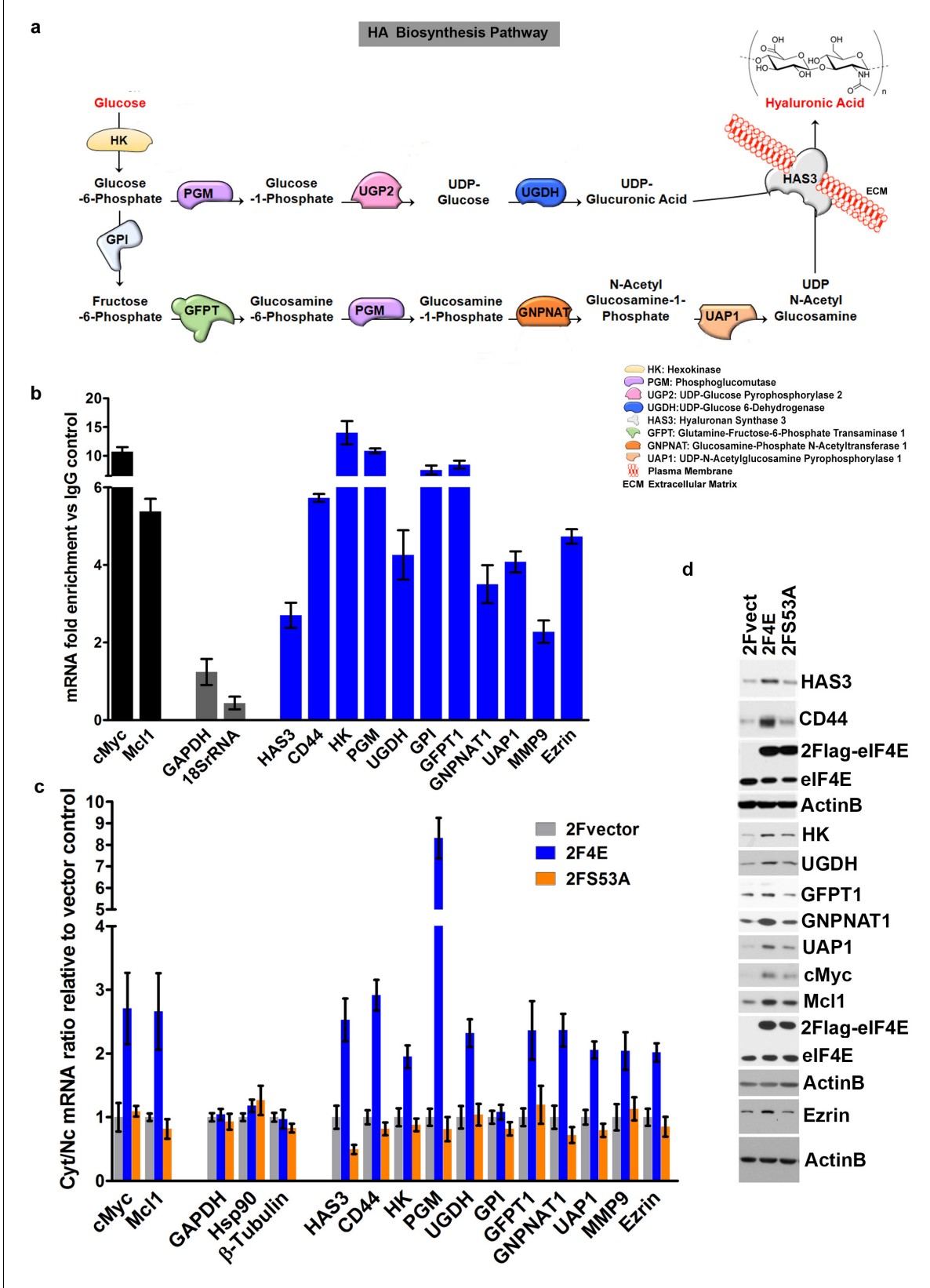

**Figure 1.** eIF4E regulates the expression of HA synthesizing enzymes and HA receptor CD44. (**A**) HA biosynthesis pathway. (**B**) RT-qPCR of HA synthesizing enzymes and its receptor CD44 following RNA immunoprecipitation (RIP) using anti-eIF4E antibody from nuclei of U2Os 2Flag-eIF4E cells.

*Figure 1 continued on next page*

*Figure 1 continued*

Data are normalized to IgG control and presented as fold change. c-Myc and Mcl-1 are positive targets of eIF4E and thus serve as positive controls, whereas GAPDH and 18S rRNA served as negative controls. We used standard deviation to denote statistical significance. One representative experiment is shown, which was carried out in triplicate. (C) RT-qPCR of HA synthesizing enzymes in cytoplasmic versus nuclear fractions of U2Os cells overexpressing 2Flag-eIF4E (2F4E), S53A mutant (2FS53A) or vector control (2Fvect). Data are normalized to vector control. c-Myc and Mcl-1 served as known eIF4E targets, whereas GAPDH, Hsp90 and β-Tubulin served as negative controls. One representative experiment is shown, which was carried out in triplicate. (D) Western blot of HA synthesizing enzymes and CD44 as a function of eIF4E or S53A mutant overexpression. Mcl-1 served as positive eIF4E target control. Actin was used as a loading control. Each Actin blot corresponds to the western blots above it. Both 2Flag-eIF4E and endogenous eIF4E are shown. HK: Hexokinase; HAS3: Hyaluronan Synthase 3; PGM5: Phosphoglucomutase 5; UGP2: UDP glucose pyrophosphorylase; UGDH: UDP glucose dehydrogenase; GFPT1: Glutamine fructose 6 phospho transaminase; GNPNAT1: Glucosamine phosphate N-acetyltransferase; UAP1: UDP N-acetyl pyrophosphorylase; GPI: Glucose-6-phosphate isomerase; CD44: HA receptor; MMP9: Matrix Metalloproteinase 9. For bar graphs, the mean ± standard deviation are shown. Experiments were carried out in triplicate, at least three independent times.

DOI: https://doi.org/10.7554/eLife.29830.002

The following figure supplements are available for figure 1:

**Figure supplement 1.** eIF4E regulates HA synthesis.
DOI: https://doi.org/10.7554/eLife.29830.003
**Figure supplement 2.** eIF4E regulates HA synthesis.
DOI: https://doi.org/10.7554/eLife.29830.004
**Figure supplement 3.** eIF4E regulates HA synthesis.
DOI: https://doi.org/10.7554/eLife.29830.005

results were compared to IgG controls (*Figure 1b* and *Figure 1—figure supplement 1a,b*). To ensure these interactions were functional, we also monitored the mRNA export of candidate transcripts as a function of eIF4E overexpression by monitoring RNA content in nuclear and cytoplasmic compartments. In both experiments, RNAs were detected by quantitative reverse transcription PCR methods (RT-qPCR). Fractionation quality was assessed using U6snRNA and tRNA$^{lys}$ for nuclear and cytoplasmic fractions, respectively. Preliminary studies using genome-wide screens of our nuclear eIF4E RIPs provided evidence that factors involved in HA biosynthesis were targets (*Figure 1a*). Using RT-qPCR, we determined that the transcripts encoding most of these enzymes bound eIF4E in nuclear RIPs with enrichments ranging from 2.5 to 15 fold (*Figure 1b*). These target mRNAs include hyaluronan synthase 3 (HAS; 2.6 fold), as well as many of the enzymes involved in generating the UDP-Glucuronic acid and UDP-N-Acetyl Glucosamine precursors including hexokinase 1 (HK, ~15 fold), and phosphoglucomutase (PGM5, ~11 fold), amongst others (*Figure 1b*). Comparison of eIF4E RIPs to input gave a similar pattern of results as to those compared to IgG controls (*Figure 1—figure supplement 2a*). We note that comparisons to input for abundant RNAs can lead to false negatives because even if fully bound to eIF4E, levels of nuclear eIF4E may not be sufficient to deplete the input pool. Further comparison to IgG allows an assessment of background binding to beads.

Nuclear mRNA export assays indicated that these mRNA-eIF4E interactions were functional since we observed increased mRNA export two- to eight-fold upon eIF4E overexpression relative to vector controls depending on the transcript monitored (*Figure 1c*). The only transcript which was bound to eIF4E in the nucleus but was not an export target in these cells encoded glucose phosphoisomerase (GPI). However, many isoforms of GPI exist and thus future studies exploring specific isoforms may identify this as being actively exported. In addition to the HA biosynthetic machinery, other eIF4E targets identified in our RIP and fractionation studies included downstream effectors of HA for example CD44 as well as its signaling partners for example Ezrin and MMP9 (*Figure 1b and c*) (*Montgomery et al., 2012*). Total levels for target mRNAs were not affected by eIF4E overexpression confirming that these effects were post-transcriptional (*Figure 1—figure supplement 1c*). Negative control RNAs such as GADPH, Hsp90 and β-Tubulin were not in the immunoprecipitations and were not modulated at the export level (*Figure 1b,c*). For comparison, we used the S53A eIF4E mutant which is deficient in mRNA export and transformation but active in translation (*Osborne and Borden, 2015*; *Culjkovic-Kraljacic et al., 2012*). As expected, the S53A eIF4E mutant did not promote export of any of these mRNAs (*Figure 1c*). We also assessed whether the HA pathways transcripts were targets of eIF4E at the translation level using polysomal analysis in eIF4E-overepxressing or vector control U2Os cells and monitored mRNAs by RT-qPCR in each fraction. We did not observe substantial shifts for most of these mRNAs on the polysomes with modest shifts to

heavier polysomes including CD44, UAP, UGDH and UGP2 (*Figure 1—figure supplement 3a,b*). c-Myc mRNA, an established translational target of eIF4E, showed a substantial polysomal shift (*Figure 1—figure supplement 3a,b*) (*Culjkovic-Kraljacic et al., 2016*). Our previous studies demonstrated that whether mRNAs were translation targets of eIF4E could be cell-type dependent (*Culjkovic-Kraljacic et al., 2016*), and thus, while most HA pathways transcripts do not appear to be substantially shifted on polysomes in U2Os cells, it is possible that the HA-pathway may be more translationally regulated by eIF4E in other cell types. In U2Os cells, the effects of eIF4E on the HA pathway appear to be more dependent on its mRNA export functions.

Consistent with our above findings, eIF4E overexpression led to increased protein levels of the relevant enzymes, including HAS3, HK, GFPT1, GNPNAT1, UAP1, UGDH, as well as downstream effectors of HA function such as CD44 and Ezrin, relative to vector controls (*Figure 1d*). Available antibodies for PGM5, UGP2 and GPI were not of sufficient quality to assess their respective protein levels. The positive controls Mcl1 and c-Myc were elevated in eIF4E overexpressing cells relative to vector controls, as expected (*Figure 1d*). Although the S53A eIF4E mutant did not export the corresponding transcripts, a modest increase relative to vector controls was observed at the protein levels for UAP1, c-Myc and Mcl-1 consistent with modest shifts in the polysomal profiles for these transcripts (*Figure 1d* and *Figure 1—figure supplement 3*) (*Culjkovic-Kraljacic et al., 2016*; *Rousseau et al., 1996*). Notably, the changes induced by S53A eIF4E were substantially less than those observed for wild-type eIF4E for all cases, consistent with the role of RNA export in their expression. Thus, eIF4E increases levels of the enzymes in the HA biosynthetic pathway and downstream effectors of HA signalling (e.g. CD44 and Ezrin) through increased mRNA export.

Next, we determined whether eIF4E overexpression drove production of HA. To address this, we monitored HA levels in eIF4E, eIF4E S53A and vector control U2OS cells using biotinylated HA-binding protein (HABP) with streptavidin-FITC and confocal microscopy. HA levels were substantially elevated in eIF4E overexpressing cells relative to vector controls or S53A eIF4E cells *Figure 2a*). Strikingly, HA was not extruded into the media, but rather coated the cell surface and formed short, filamentous protrusions radiating from the surface coat in the eIF4E overexpressing cells (*Figure 2a, b*). Enzymatic depletion of HA with *Streptomyces* hyaluronidase (HAse) virtually eliminated the HA signal indicating that the staining was specific and suggesting that the structures were HA-dependent (*Figure 2a*). Our findings are consistent with studies which used HAS3 overexpression to artificially induce HA production (1) where the protrusions were too narrow (120–130 nm) to be seen by light microscopy but were readily detectable using fluorescent HABP conjugates. We used fluorescence-assisted carbohydrate electrophoresis (FACE) to independently validate elevated HA production (*Figure 2c* and *Figure 1—figure supplement 1e*). We observe a ~ threefold increase in HA levels in eIF4E-overexpressing cells relative to vector controls. HA levels in S53A-eIF4E cells were much lower than eIF4E overexpressing cells, and only modestly elevated relative to vector controls consistent with the mutant's modest effects on the HA biosynthetic enzymes. Further, removal of extracellular glucose led to reduction of HA signalling consistent with the use of glucose as the major metabolic precursor in this pathway (*Figure 1—figure supplement 1g–h*). Thus, eIF4E overexpression induced HA production and was found associated with cells, coating the surface and forming protrusions. eIF4E required its mRNA export activity for HA production and this was likely augmented by its translation activity.

We hypothesized that HA levels would be repressed by inhibition of eIF4E. eIF4E-overexpressing cells were treated with either RNAi to eIF4E or with a pharmacological inhibitor, ribavirin (*Figure 2c, d*). Ribavirin directly binds eIF4E and inhibits its mRNA export and translation functions (*Pettersson et al., 2015*; *Kentsis et al., 2004*; *Volpon et al., 2013*). We observed a reduction in HA to background levels via confocal microscopy using either ribavirin treatment or RNAi knockdown of eIF4E. Using FACE, we similarly observed a ~ ninefold reduction in HA levels for both eIF4E knockdown relative to control RNAi and ~2.5-fold for ribavirin treated versus untreated cells (*Figure 2c* and *Figure 1—figure supplement 1f*). Thus, eIF4E is necessary for HA production in these cells.

We extended our studies to assess whether eIF4E drives HA production in cellular contexts characterized by naturally occurring elevation of eIF4E for example acute myeloid leukemia (AML) and breast cancer (*Assouline et al., 2015*; *Pettersson et al., 2015*; *Assouline et al., 2009*; *Pettersson et al., 2011*). First, we examined the MM6 AML cell line which is characterized by elevated nuclear eIF4E levels, and thus with increased mRNA export activity for eIF4E targets (*Figure 3a–e* and *Figure 3—figure supplement 1a–d*). Using nuclear RIPs and mRNA export assays,

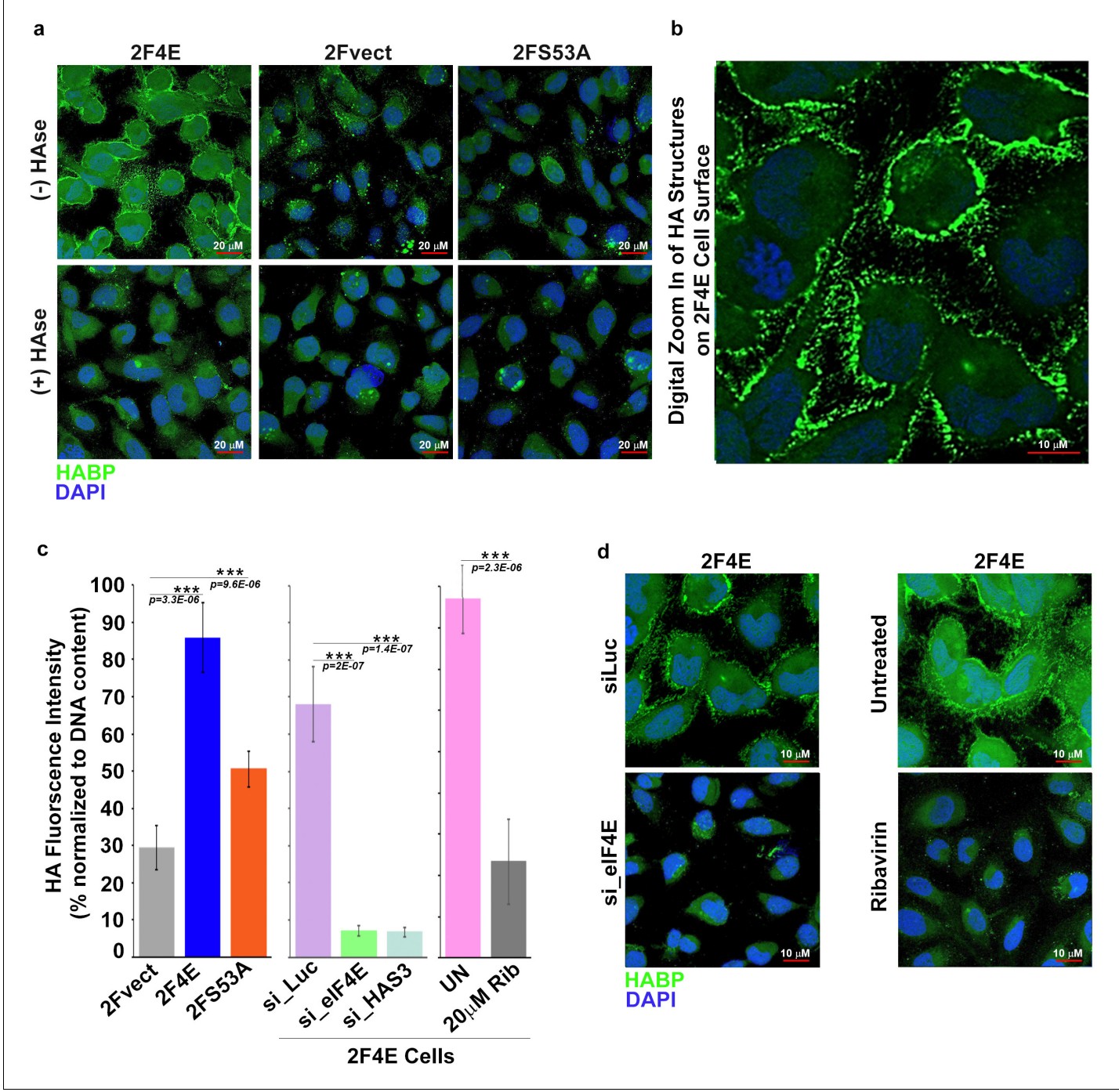

**Figure 2.** eIF4E overexpression correlates with increased HA synthesis. (A) Fluorescence staining of HA (in green) using biotinylated HA-binding protein with streptavidin-FITC in U2Os cells overexpressing eIF4E, S53A mutant or vector control in the presence or absence of *Streptomyces* Hyaluronidase treatment. DAPI is in blue. Note cell surface expression of HA in response to eIF4E overexpression. All confocal settings are identical between specimens and thus lower signal is indicative of less HA. A × 40 objective with no digital zoom was used. (B) 2x digital zoom in confocal images of HA from part (A). (C) Quantification of fluorophore-assisted carbohydrate electrophoresis (FACE) gels (Sup *Figure 1e&f*) for HA levels in U2Os cells expressing eIF4E, S53A mutant or vector control, and U2Os cells overexpressing eIF4E following HAS3/eIF4E knockdown or pharmacological inhibition with ribavirin (Rib). (D) Fluorescence staining of HA (in green) following siRNA to eIF4E or ribavirin treatment in U2Os cells overexpressing eIF4E. DAPI is in blue. A × 63 objective with no digital zoom used. For bar graphs, the mean ± SD are shown. Experiments were carried out in triplicate, at least three independent times. **p < 0.01, ***p < 0.001 (Student's t-test).

DOI: https://doi.org/10.7554/eLife.29830.006

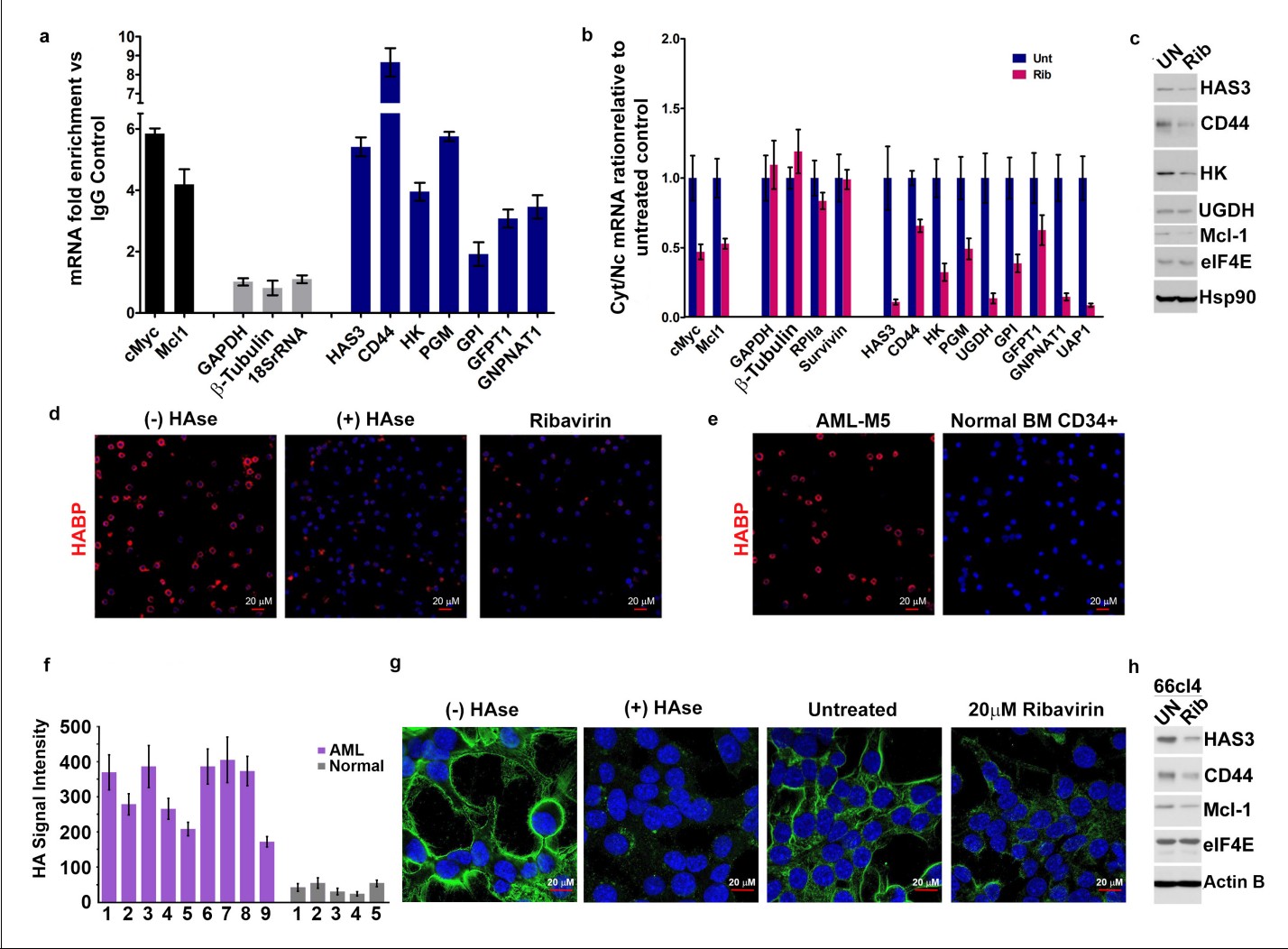

**Figure 3.** eIF4E elevates HA in cancer cell lines and primary specimens. (**A**) RT-qPCR of HA synthesizing enzymes and its receptor CD44 following RNA immunoprecipitation (RIP) from nuclei of MM-6 cells using anti-eIF4E antibody. Data are presented as fold change relative to IgG controls. c-Myc and Mcl-1 served as endogenous eIF4E targets, while GAPDH, β-Tubulin and 18SrRNA served as a negative control. (**B**) RT-qPCR of HA synthesizing enzymes in cytoplasmic versus nuclear fractions of MM-6 cells treated with Ribavirin (Rib). Data are normalized to untreated control. Error bars are means ± S.D. c-Myc and Mcl-1 served as positive controls since they are known eIF4E targets, while GAPDH, β-Tubulin, RPIIa and Survivin served as negative controls. (**C**) Western blot of HA synthesizing enzymes and CD44 as a function of Ribavirin treatment in MM-6 cell line. Mcl-1 served as endogenous eIF4E target, while Hsp90 served as a loading control. (**D**) Fluorescence staining of HA (in red) in MM-6 cell lines treated with Ribavirin (Rib) versus untreated (UN) in the presence or absence of Hyaluronidase treatment. DAPI is in blue. A 63x objective with no digital zoom was used. (**E**) Fluorescence staining of HA (in red) in human CD34+ specimens from healthy volunteer compared with leukemic cells from M5 AML Patient. (**F**) Quantification of HA fluorescence staining in 9 M4/M5 AML patients and CD34+ specimens from five healthy volunteers using ZEN software. HA signal intensity is presented as the geometric means of the HA signal. The mean ± standard deviations are shown. (**G**) Fluorescence staining of HA (in green) in 66cl4 cells in the presence or absence of Hyaluronidase or Ribavirin treatment. DAPI is in blue. A × 63 objective with no digital zoom used. (**H**) Western blot control of HAS3 and CD44 as a function of ribavirin treatment in 66cl4 cell line. Mcl-1 served as endogenous eIF4E target, while ActinB served as loading control. Experiments were carried out in triplicate, at least three independent times. For bar graphs, the mean ±standard deviation are shown.

DOI: https://doi.org/10.7554/eLife.29830.007

The following figure supplements are available for figure 3:

**Figure supplement 1.** eIF4E elevates HA in cancer cell lines and primary specimens.
DOI: https://doi.org/10.7554/eLife.29830.008
**Figure supplement 2.** eIF4E concentrates in HA rich protrusions and correlates with sites of active translation.
DOI: https://doi.org/10.7554/eLife.29830.009

we found that all mRNAs for the HA biosynthesis machinery including HAS3 and CD44 are eIF4E export targets in this cell type (*Figure 3a–c*). These targets included transcripts encoding GPI, which was not an export target in U2Os cells. This suggests that the ability to promote HA production in these cells might be even more potent and also that the cell context plays a role particularly in terms of isoform content of RNAs and protein compliment. We also note diversity in terms of the enzyme family members associated with eIF4E in MM6 cells versus eIF4E-overexpressing U2Os cells. For instance, transcripts encoding PGM5 which were eIF4E targets in U2Os cells, were not well expressed in MM6 cells. Instead, eIF4E bound to and exported PGM1 mRNAs. Importantly, these conservative substitutions in enzyme content still led to increased HA biosynthesis as observed by FACE and HABP staining (*Figure 3d*). Similar to U2Os cells, the surface of MM6 cells was characterized by HA coats. We also found in MM6 cells that eIF4E targeting with ribavirin reduced the mRNA export of the corresponding HA enzymes and CD44 from two- to ninefold depending on the mRNA monitored (*Figure 3b*). Ribavirin did not alter total mRNA levels consistent with this being a post-transcriptional effect (*Figure 3—figure supplement 1c*). Consistently, ribavirin treatment dramatically lowered protein levels for the representative members of the HA biosynthetic machinery examined: HAS3, HK, UGDH as well as CD44 (*Figure 3c*). HA staining was reduced to background levels by ribavirin or HAse treatment as observed by HABP staining and confocal microscopy (*Figure 3d*). We paralleled these studies in primary high-eIF4E AML specimens and also in CD34+ bone marrow specimens isolated from healthy volunteers (*Figure 3e–f*). Consistent with the MM6 cells, 9/9 high-eIF4E AML specimens had HA coats and protrusions with HA staining 4–10 fold higher than the five normal human CD34+ specimens which all showed low-level, presumably background, HA staining similar to intensities observed in vector controls U2Os cells (*Figure 2a*). These findings suggest that the HA coat is specific to the malignant state and not a general feature of blood cells (*Figure 3e–f*). These findings point to new functionalities for HA in leukemia cells, where it was previously thought HA only played a role in the bone marrow stroma and not on the leukemia cells themselves.

Next, we examined another high-eIF4E context, the breast cancer cell line 66cl4. These cells had highly elevated nuclear eIF4E, and also had readily visible HA-surface coats and protrusions as observed by confocal microscopy (*Figure 3g*). Ribavirin treatment reduced HAS3 and CD44 levels relative to untreated controls as well as dramatically reduced HA levels (*Figure 3g*). Furthermore, we observed by in situ translation studies using fluorescence non-canonical amino-acid tagging (FUN-CAT), that there could be active translation down the length of the protrusions in eIF4E-overexpressing and 66cl4 cells (*Figure 3—figure supplement 2a–b*). These translation foci are cyclohexamide dependent validating them as sites of ongoing translation (*Figure 3—figure supplement 2a–b*). We postulate that eIF4E could be involved in localized protein synthesis to spatially couple translation of relevant HA enzymes with HA biosynthesis. In this way, increased rates of translation may not be required, but localization of translation could facilitate HA synthesis as well as mRNA export. In all, our studies demonstrate that eIF4E controls HA biosynthesis at both the mRNA export and translation level thereby coordinately driving HA production.

eIF4E plays well-established roles in invasion, migration and metastasis. We hypothesized that eIF4E co-opted HA synthesis to execute these activities. Starting with invasion, we observed that eIF4E-overexpressing cells invaded matrigel fourfold better than vector controls (*Figure 4a,b* and *Figure 4—figure supplement 1a–b and e–f*). By contrast, the S53A eIF4E mutant increased invasion by only 50% relative to vector controls consistent with its more modest effects on HA production by FACE (*Figure 2c*; *Figure 4—figure supplement 1e–f* and *Figure 1—figure supplement 1e*). In both eIF4E overexpressing and vector control cells, RNAi to eIF4E or ribavirin treatment reduced invasion by ~2.5-fold consistent with the significant reduction in HA levels (*Figure 2c*, *Figure 4e–f* and *Figure 4—figure supplement 1e–f*). To determine the relevance of HA production specifically to eIF4E mediated invasion, we used RNAi to knockdown HAS3 in eIF4E overexpressing and vector cells (*Figure 4a* and *Figure 4—figure supplement 1a–b*). We focused on HAS3 since this is the last step in the biosynthetic pathway of HA and its inhibition specifically impairs HA synthesis, whereas other enzymes in this pathway also participate in unrelated processes. We observed that RNAi to HAS3 reduced the invasion activity of eIF4E by ~ fivefold in eIF4E overexpressing cells (to levels of RNAi controls) and 2.5-fold in vector controls where the effects of endogenous eIF4E are likely being targeted. HAS3 knockdown did not affect eIF4E levels as observed by western blot (*Figure 4d*). Furthermore, confocal microscopy experiments revealed that RNAi to HAS3 decreased HA levels to background consistent with its baseline invasion activity (*Figure 4c*). FACE studies also revealed that

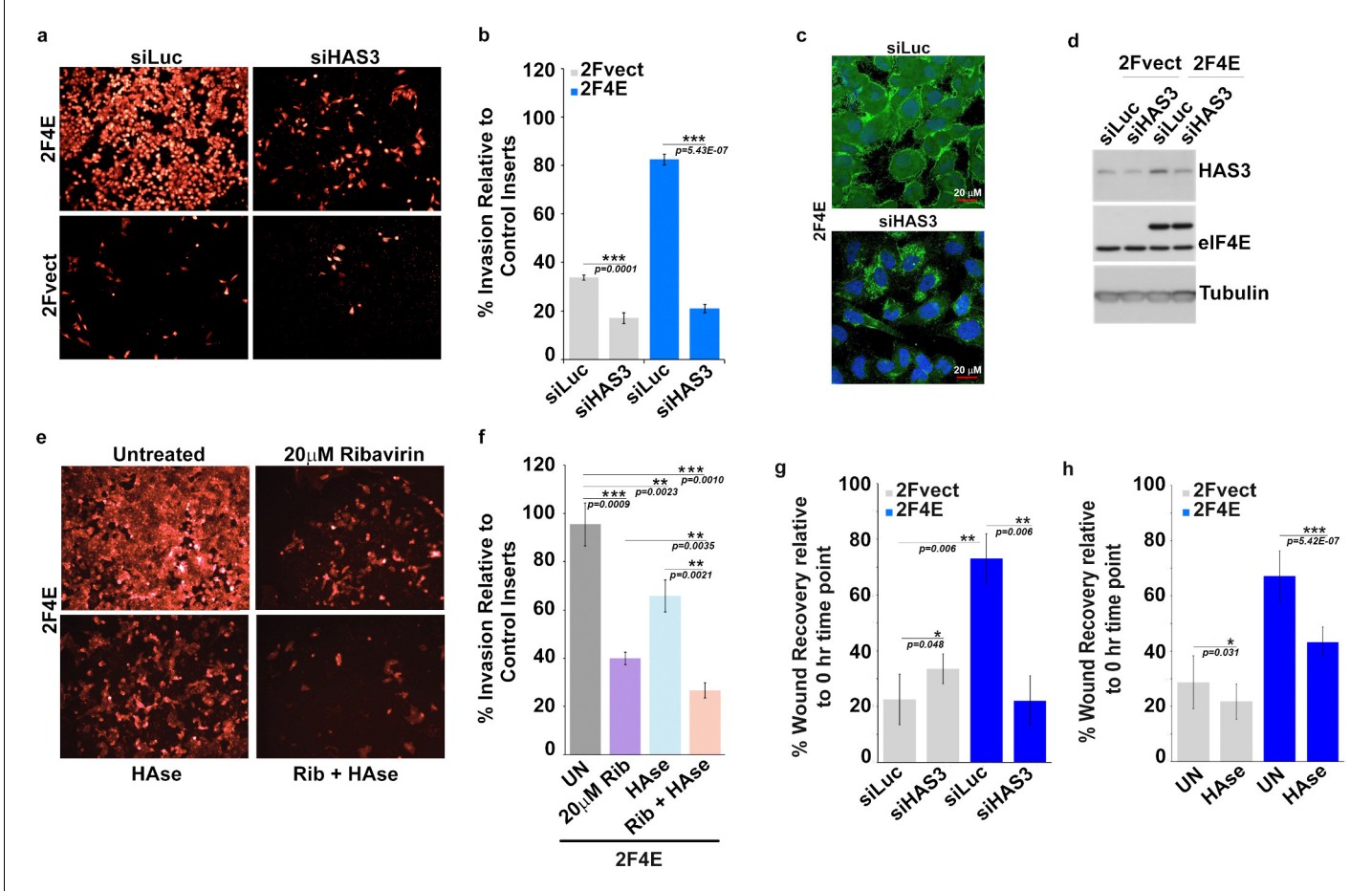

**Figure 4.** Surface HA is required for eIF4E-mediated invasion and migration of cancer cells. (A–B) Cell invasion through matrigel assessed following siRNA-mediated knockdown of HAS3 or scrambled control (Luciferase (siLuc)) in U2Os cells overexpressing eIF4E (2F4E) or vector (2FVect). Invasion is measured as the percentage of fluorescence staining intensity in matrigel coated inserts versus that of the control inserts. (C) Fluorescence staining of HA (in green) following siRNA to HAS3. DAPI is in blue. A × 40 objective with no digital zoom used. (D) Western blot to demonstrate knockdown efficiency of HAS3. Tubulin served as loading control. (E–F) Cell invasion through matrigel assessed in U2Os cells overexpressing eIF4E following treatment with Ribavirin (Rib) and/or Hyaluronidase. Invasion is measured as the percentage of fluorescence staining intensity in matrigel-coated inserts versus that of the control inserts. (G–H) Cell migration across a scratch assessed in U2Os cells overexpression eIF4E or vector control following knockdown of HAS3 or treatment with *Streptomyces* Hyaluronidase. Migration is measured as the percentage of the area not filled with cells at t = 16 hr normalized to that of the t = 0 hr time point. For bar graphs, the mean ±standard deviation are shown. Experiments were carried out in triplicate, at least three independent times. **$p < 0.01$, ***$p < 0.001$ (Student's t-test).

DOI: https://doi.org/10.7554/eLife.29830.010

The following figure supplements are available for figure 4:

**Figure supplement 1.** Surface HA is required for eIF4E-mediated invasiveness of cancer cells.
DOI: https://doi.org/10.7554/eLife.29830.011

**Figure supplement 2.** CD44 is required for the invasion of eIF4E cells.
DOI: https://doi.org/10.7554/eLife.29830.012

HAS3 knockdown lowered HA levels by ~ ninefold relative to RNAi controls (*Figure 2c* and *Figure 1—figure supplement 1f*). Strikingly, eIF4E knockdown and HAS3 knockdown similarly reduced HA to background levels.

We explored the efficacy of HAse treatment for the invasion activity of eIF4E (*Figure 4e and f*). Given the length of time between HAse treatment and re-emergence of HA in eIF4E-overexpressing cells was 12 hr (*Figure 4—figure supplement 1h*), we treated cells with HAse every 8 hr to ensure HA was depleted during the course of the experiments. We observed that HAse treatment reduced invasion by 40% relative to untreated controls. For comparison, ribavirin decreased invasion by 60%

consistent with its reduction of HA to background levels. Strikingly, the combination of ribavirin and HAse reduced invasion by 80% (*Figure 4e–f* and *Figure 4—figure supplement 1c–d*). Importantly, ribavirin affects multiple eIF4E target pathways, not only HA biosynthesis, and thus its effects are expected to be greater than HAse alone. We extended these studies to monitor the role HA production played in the migration activity of eIF4E in wound healing assays. As expected, we observed increased migration in eIF4E-overexpressing cells relative to vector controls (*Figure 4g,h*). Knock-down of HAS3 reduced eIF4E-dependent migration by ~ fourfold while treatment with HAse reduced it by almost twofold indicating this was also an HA dependent phenomenon.

We extended the above studies to examine the role of HA in eIF4E-mediated metastasis in vivo. Previously, we demonstrated that ribavirin treatment reduced metastasis by 3-fold in an eIF4E-dependent pulmonary metastatic mouse model using 66cl4 cells (*Pettersson et al., 2015*). In this previous study, 66cl4 cells were injected into the mammary fat pad of syngeneic Balb/c mice. On day 11 post injection, mice with palpable tumors were randomized into two groups and treated orally with water or 3 mg ribavirin/mouse/day (5d/week, to reach, clinically achievable, plasma concentration of 23 μM). On day 27, all mice were sacrificed and tumors and lungs were preserved. Metastatic lung burden was calculated as the percentage of tumor/total lung areas for each mouse on 5 × 50 μm serial step sections by manually circling the metastasis (Visiomorph Software). Above, we demonstrated that eIF4E drives HA production in these cells and that ribavirin repressed HAS3 and HA production (*Figure 3*). Here, we investigated these tissue sections to ascertain whether inhibition of eIF4E activity correlated not only with reduced lung metastases (*Pettersson et al., 2015*) but also with lower HA levels (*Figure 5*). Serial formalin-fixed sections of tumor bearing lungs, from control and ribavirin-treated animals, were stained for HA, and hematoxylin (*Figure 5a*). Staining intensity and area were quantified using Visiomorph. Tumors from 10 animals for treated and nine in the

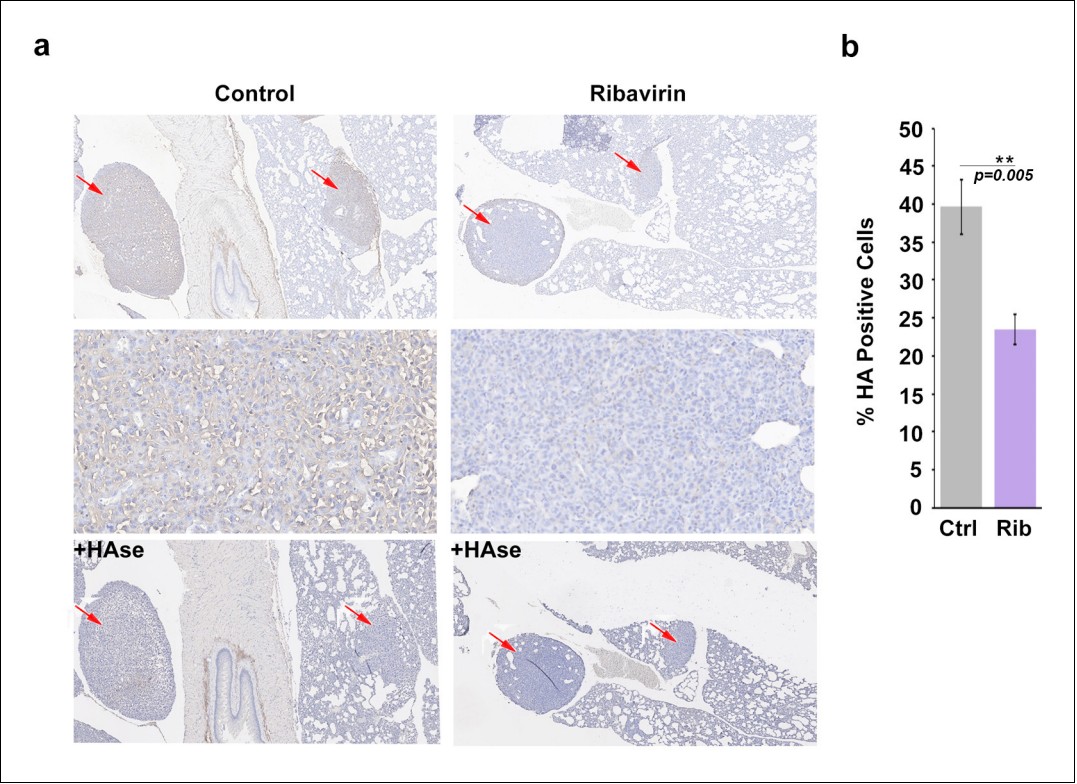

**Figure 5.** HA biosynthesis is required for eIF4E-mediated lung metastasis in mice. (A) Histochemical staining of HA using biotinylated HABP in metastatic mouse tumors. +HAse indicates sections were treated with HAse prior to biotinylated HABP to ensure HA staining was specific. Red arrows indicate tumors. 3X (first and third row) and 50X (second row) magnification are presented. (B) Quantification from Visiomorph for all 19 mice over multiple sections per animal. For bar graphs, the mean ± standard deviation are shown. *p < 0.05 (Student's t-test).
DOI: https://doi.org/10.7554/eLife.29830.013

untreated groups were analyzed. Strikingly, we observed a 50% decrease in HA levels in the lungs of ribavirin-treated mice relative to controls (*Figure 5a–b*). HA surrounded tumor cells indicating that HA was adjacent to and/or on the surface of these cells. As expected, HA was also found in normal tissues, consistent with its major structural role in the microenvironment. HAse treatment of serial sections indicated that the HA staining observed is specific (*Figure 5a*). Thus, we demonstrate that eIF4E targeting leads to reduced HA levels and decreased metastasis in vivo.

Next, we examined whether CD44, as a representative downstream effector of the HA-network, was also required for eIF4E-mediated invasion (*Figure 1b–d*). eIF4E overexpression led to highly elevated CD44 protein levels (*Figure 1d*) and CD44 coated the surface (*Figure 4—figure supplement 2a*). Indeed, CD44 and HA co-localized on the cell surface (*Figure 4—figure supplement 2a and e*). These studies are similar to those showing that CD44 bound HA on the surface of HAS3 overexpressing cells (*Kultti et al., 2006*). Inhibition of CD44 with RNAi or independently with CD44-blocking antibodies (mAb A3D8) reduced eIF4E-mediated invasion by ~ fourfold for RNAi and ~ threefold for mAb A3D8 treatment (*Figure 4—figure supplement 2b–c and f–g*). The similarity for the reduction induced by CD44 inhibition, knockdown of HAS3 or ribavirin treatment suggests that CD44 plays an important role in this process. Significantly, modulation of CD44 did not alter HA status, as seen by HABP-staining via confocal microscopy (*Figure 4—figure supplement 2e*). Thus, although HA is essential for eIF4E-mediated invasion, its co-factors CD44 and perhaps others, are likely also required for this activity. This suggests that the HA coat and/or protrusions need to be armed with CD44 to support the invasion potential of eIF4E overexpressing cells. Further in high-eIF4E cells, HA is likely directly binding CD44 to activate this signaling pathways in neighboring cells and/or in its own cells leading to autocrine stimulation. In either case, these would stimulate cellular signaling cascades that drive the oncogenic phenotype.

## Conclusions

Our studies demonstrate that the entire HA-network is subjected to coordinated post-transcriptional control by eIF4E, in effect, an RNA regulon controlling HA biosynthesis and ultimately cell surface architecture. Our data suggest that HA levels would scale with both elevation and loss of control of eIF4E. Thus, in normal cells with less eIF4E and tighter eIF4E regulation, HA biosynthesis would be expected to be lower. In cancer cells, elevation of eIF4E and loss of key regulators of eIF4E such as PRH/Hex in AML (*Topisirovic et al., 2003a*; *Topisirovic et al., 2003b*), allows more of these transcripts to be bound to eIF4E in the nucleus, have their export elevated and in some cases be better translated. These findings have several mechanistic and clinical implications. For instance, factors that modulate the levels, localization, or phosphorylation of eIF4E are positioned to profoundly affect HA production and the global activity of the HA network. Indeed, targeting eIF4E with RNAi knockdown or ribavirin treatment reduced HA levels as effectively as direct targeting with HAse. We note that eIF4E drove the production of cell-associated HA which in turn, fundamentally modified the cell-surface architecture facilitating invasion and metastases. Thus eIF4E may specifically drive an oncogenic HA-programme, in contrast to typical situations where this large glycosaminoglycan is extruded into the matrix. It is not yet known if the ability to modulate HA is a property unique to eIF4E, or if other oncoproteins initiate a similar programme.

## Materials and methods

### Reagents and constructs

pcDNA-2Flag-eIF4E wild-type and S53A mutant constructs were previously described (*Culjkovic et al., 2005*; *2006*; *Culjkovic-Kraljacic and Borden, 2013*). Ribavirin was purchased from Kemprotec, UK (CAS 36791-04-5). AMAC (2-aminoacridone) from Molecular probes, CA(Cat# A-6289). Sodium Cyanoborohydride from Sigma Aldrich, CA (Cat # 15,615-9). Chondroitinase ABC from Proteus Vulgaris from Sigma Aldrich (Cat# C3667-10UN). 40% Acrylamide/Bis Solution, 37.5:1 from BioRad technologies, CA (Cat# 1610148).

Antibodies for immunoblotting: Mouse monoclonal anti-eIF4E (BD PharMingen, CA, Cat# 610270) or Rabbit monoclonal anti-eIF4E (Millipore, CA, Cat# 04–347) for Western Blot analysis. Rabbit anti-eIF4E for RNA immunoprecipitation (MBL international, MA, U.S.A, Cat# RN001P). Mouse monoclonal anti-CD44 blocking antibody A3D8 (Novus Biologicals, CA, Cat# NB600-1457).

Mouse monoclonal anti-CD44 antibody (156–3 C11) for Western Blot and Confocal Analysis (Cell Signalling, CA, Cat# 3570). Rabbit polyclonal anti-HAS3 antibody (Abcam, CA, Cat# ab154104). Rabbit polyclonal anti-phosphoglucomutase 5 (Abgent, CA, U.S.A, Cat# AI14638). Rabbit polyclonal anti-Glucose six phosphate isomerase antibody [EPR11663(B)] (Abgent, AW5240-U400). Rabbit polyclonal anti-UDP glucose dehydrogenase antibody (Abgent, AP12613b). Mouse monoclonal anti-GFPT1 antibody [EPR4854] (Abgent, AO2212a). Rabbit polyclonal anti-GNPNAT1 antibody (GeneTex, CA, U.S.A, Cat# GTX122246). Rabbit monoclonal anti-UAP1 antibody [EPR10259] (Abcam, ab155287). Mouse monoclonal anti-β-actin (Sigma Aldrich, A5441). Mouse monoclonal anti-α-tubulin (Sigma Aldrich, T5168). Rabbit polyclonal anti-Mcl-I (S-19) (Santa Cruz, TX, U.S.A, Cat# sc-819). Mouse monoclonal anti-c-Myc (9E10) (Santa Cruz, sc-40). Mouse monoclonal anti-HSP90α/β (F-8) (Santa Cruz, sc-13119). Rabbit polyclonal anti-Lamin A (C-terminal) (Sigma Aldrich, L1293). Rabbit polyclonal anti-Pol II N-20 (Santa Cruz, sc-889). Rabbit polyclonal anti-GAPDH (FL-335) (Santa Cruz, sc-25778).

## Cell culture and transfection

U2Os cells (obtained from ATCC, CA) were maintained in 5% $CO_2$ at 37°C in Dulbecco's modified Eagle's medium (DMEM) (ThermoFisher Scientific, CA, Cat# 11995–065) supplemented with 10% fetal bovine serum (FBS) (ThermoFisher Scientific, Cat# 12483–020) and 1% penicillin-streptomycin (ThermoFisher Scientific, Cat# 15140122). Mono-Mac-6 (MM6) cells (obtained from DSMZ, Cat# ACC 124) were maintained in RPMI 1640 (ThermoFisher Scientific) supplemented with 10% FBS, 1% penicillin-streptavidin, 1% MEM non-essential amino acids (ThermoFisher Scientific, Cat# 11140076) and 10 µg/ml recombinant human insulin (Sigma Aldrich, Cat# 91077C). Mouse mammary 66cl4 cells were obtained from Dr. Josie Ursini-Siegel (Lady Davis Institute, Montreal, QC, Canada) and cultured in RPMI 1640 (ThermoFisher Scientific, Cat# 22400–089) supplemented with 10% heat-inactivated FBS and 1% penicillin-streptomycin. The identity of U2Os and MM6 cell lines has been authenticated using STR profiling (Montreal EpiTerapia Inc.). However, for the 66cl4 mouse cell line there are no universal standards for authentication. All cell lines were routinely checked to ensure that there was no mycoplasma contamination using MycoAlert Mycoplasma Detection kit (Lonza,NY, U.S.A, Cat# LT07-418). Transfections for stable cell lines were performed using TransIT-LT1 Transfection Reagent (Mirus, CA, Cat# MIR 2300) as specified by the manufacturer, and selected in G-418-containing medium (1 mg/mL) for eIF4E stable overexpressing cell lines (G418-Sulfate was obtained from Wisent Bioproducts, CA, Cat# 400–130-IG). For eIF4E, HAS3 or CD44 knockdowns, U2Os cells were transfected with Lipofectamine 2000 (ThermoFisher Scientific, Cat# 11668019) and 20–40 nM siRNA duplex according to the manufacturer's instructions. For siHAS3 two sequences were used such that the total amount of the siRNA mix is equal to 40 nM. Cells were analyzed 96 hr after transfection. List of siRNA purchased:

| siRNA | Catalog # |
|---|---|
| siRNA duplex_eIF4E (Mouse) | IDT Technologies, CA<br>MMU.RNAI.N0007917.1.1<br>MMU.RNAI.N007917.1.2 |
| siRNA duplex _eIF4E (Human) | IDT Technologies, CA<br>sense (CUGCGUCAAGCAAUCGAGAUUUGGG)<br>antisense (CCCAAAUCUCGAUUGCUUGACGCAGUC) |
| siRNA duplex _HAS3 | IDT Technologies, CA<br>HSC.RNAI.N138612.12.5<br>HSC.RNAI.N005329.12.3 |
| siRNA duplex _CD44 | Qiagen, CA<br>FlexiTube siRNA Hs CD44 Cat# S100299705 |
| siRNA duplex _Luciferase | IDT Technologies, CA<br>sense (CACGUACGCGGAAUACUUCGAAATG)<br>antisense (CAUUUCGAAGUAUUCCGCGUACGUGUU) |

## Cellular fractionation and export assay

Fractionation protocol was followed as previously described (*Culjkovic-Kraljacic et al., 2016*). About $3 \times 10^7$ cells were collected and washed twice in ice cold PBS (1200 rpm/3–5 min) and then re-suspended with slow pipetting in 1 ml of lysis buffer B (10 mM Tris pH 8.4, 140 mM NaCl, 1.5 mM MgCl2, 0.5% NP40, 1 mM DTT and 100 U/ml RNase Inhibitors (ThermoFisher Scientific, Cat# 10777–019). The lysate was centrifuged at 1000 g for 3 min at 4°C and supernatant (cytoplasmic fraction) was transferred into a fresh microtube. The pellet (nuclear fraction) was resuspended in 1 ml of lysis buffer B, transferred to round bottom, polypropylene tube and 1/10 vol (100 µl) of detergent stock (3.3% (w/v) Sodium Deoxycholate, 6.6% (v/v) Tween 40 in DEPC H20) was added with slow vortexing (to prevent the nuclei from clumping) and incubated on ice for 5 min. The suspension was transferred to a microtube and centrifuged at 1000 g for 3 min at 4°C. Supernatant (post-nuclear fraction) was transferred to a fresh tube and the pellet-nuclear fraction was rinsed in 1 ml of lysis buffer B and centrifuged at 1000 g for 3 min at 4°C. The postnuclear and cytoplasmic fractions were combined. The RNA was extracted from the different fractions by adding TRIzol reagent (ThermoFisher Scientific, Cat# 15596026) and processed according to the manufacturer's instructions.

## RNA immunoprecipitation (RIP)

RIP from nuclear fractions of cells was performed as previously described (*Culjkovic-Kraljacic et al., 2016*). Briefly, 1 mg of nuclear lysate was used for RIP with 7 µg anti-eIF4E antibody (MBL RN001P) or control immunoglobulin G (rabbit, Millipore). After incubation, complexes were eluted by boiling in tris(hydroxymethyl)aminomethane-EDTA containing 1% sodium dodecyl sulfate and 12% β-mercaptoethanol. RNA were isolated using TRIzol reagent and isolated using Direct-zol RNA Microprep Kit (Zymo Research, CA, U.S.A, Cat# R2050).

Note that the complete list of hits from our genome-wide nuclear eIF4E RIP screens is not being provided because the screen was done only once and thus lacks statistical power. However, these results were validated by RT-qPCRs outlined in the text.

## Reverse transcription and quantitative PCR

DNAse treated RNA samples (TurboDNase, Ambion, CA, Cat# AM2238) were reversed transcribed using SuperScript VILO cDNA synthesis kit (for RIP experiments) (ThermoFisher Scientific, Cat# 11754–050) or MMLV reverse transcription (ThermoFisher Scientific, Cat# 28025013). QPCR analyses were performed using SensiFast Sybr Lo-Rox Mix (Bioline, MA, U.S.A, Cat# BIO-94020) in Applied Biosystems Viia7 thermal cycler using the relative standard curve method (Applied Biosystems User Bulletin #2). All conditions were described previously (*Culjkovic-Kraljacic et al., 2016*). Primers list includes:

| Name | Sequence |
|---|---|
| CD44 | Sense CGGCTCCTGTTAAATGGTATCT |
| | Antisense TCTGCTTTGTGGTCTGAGAAG |
| HAS3 | Sense CAGGAGGACCCTGACTACTT |
| | Antisense GTGGAAGATGTCCAGCATGTA |
| Hexokinase 1 | Sense GAAGATGGTCAGTGGCATGTA |
| | Antisense GGTGATCCGCCCTTCAAATA |
| GPI | Sense TCTATGCTCCCTCTGTGTTAGA |
| | Antisense CTCCTCCGTGGCATCTTTATT |
| UGDH | Sense GTGCCCATGCTGTTGTTATTT |
| | Antisense CGCCGTCCATCGAAGATAAA |
| UAP1 | Sense GCAGTGCTACAAGGGATCAA |
| | Antisense CCACCAGCTAGAAGAAGAACTG |

*Continued on next page*

*Continued*

| Name Sequence | |
|---|---|
| PGM5 | Sense TGATCTCCGAATCGACCTATCT |
| | Antisense ATATCCACTGGGTCCACTATCT |
| GNPNAT1 | Sense CCCAACACATCCTGGAGAAG |
| | Antisense CTCTGTTAGCTGACCCAATACC |
| GNPT1/GFAT | Sense ACTTTGATGGGTCTTCGTTACT |
| | Antisense ACAATCTGTCTCCCGTGATATG |
| Actin B | Sense GCATGGAGTCCTGTGGCATCCACG |
| | Antisense GGTGTAACGCAACTAAGTCATAG |
| GAPDH | Sense GAAGGTGAAGGTCGGAGTC |
| | Antisense GAAGATGGTGATGGGATTTC |
| MCL1 | Sense TTTCAGCGACGGCGTAACAAACTG |
| | Antisense TGGTTCGATGCAGCTTTCTTGGT |
| U6 | Sense CGCTTCGGCAGCACATATAC |
| | Antisense AAAATATGGAACGCTTCACGA |
| tRNALys | Sense GCCCGGATAGCTCAGT |
| | Antisense CGCCCAACGTGGGGC T |

## Western blot analysis

Western analysis was performed as described previously (*Culjkovic-Kraljacic et al., 2012*) with a modified lysis buffer (40 mM Hepes, pH 7.5, 120 mM NaCl, 1 mM EDTA, 10 mM β-glycerophosphate, 50 mM NaF, 0.5 µM NaVO3, and 1% [vol/vol] Triton X-100 supplemented with complete protease inhibitors [all were purchased from Sigma-Aldrich]). In addition, blots were blocked in 5% milk in TBS–Tween 20. Primary antibodies were diluted in 5% milk.

## Polysomal profiling

Polysomal profiling was done as described (*Tcherkezian et al., 2014*; *Culjkovic-Kraljacic et al., 2016*). Briefly, cells were treated with cyclohexamide (100 µg/ml, Sigma Aldrich, Cat# C7698) 10 min before harvesting and lysates were prepared using polysome lysis buffer (15 mM Tris pH 7.4, 250 mM NaCl, 15 mM MgCl2, 1% Triton X-100, 100 g/ml cyclohexamide, 1 mM DTT, 400 U/ml RNase inhibitors and protease inhibitors (Sigma Aldrich, Cat# 11697498001). Equal amounts (10 mg) of protein lysates were layered on a 20–50% linear sucrose gradient (20% and 50% sucrose solutions in 15 mM Tris pH 7.4, 15 mM MgCl2, 150 mM NaCl, 1 mM DTT, 100 µg/Ml cyclohexamide and 20 U/ml RNase inhibitors), mixed on Gradient Station IP Biocomp and centrifuged in a Beckman SW41Ti rotor at 92,000 g for 3 hr at 4˚C. Following centrifugation, polysomal fractions were collected by continuously monitoring and recording the A254 on a Gradient Station IP (Biocomp) attached to a UV-MII (GE Healthcare, CA) spectrophotometer. RNAs were isolated from polysomal fractions using TRIzol reagent. RNAs from each fraction were monitored using RT-qPCR.

## Glucose levels

U2Os cells overexpressing 2Flag-eIF4E or 2Flag-vector control were plated at 1 million cells per well of a 6-well plate. On the next day, cells were starved in DMEM + 0.5% FBS + 1% Pen/Strep for 16 hr at 37°C, 5% CO2. Following starvation, media was replaced with low-glucose DMEM (ThermoFisher Scientific, Cat# 11885–084) + 1 g/L D-(+)-Glucose (Sigma Aldrich, Cat# G8644) + 10% FBS + 1% Pen/Strep. Glucose levels were measured at indicated time points using Clarity Plus Blood Glucose Monitoring Kit (Cat# DTG-GL-15PROMO).

## HA synthesis growing conditions per cell line

(A)For U2Os or 66cl4 cell lines: Cells were seeded at a density of 80,000 cell per well of four-well glass Millicell EZ-Slide (Millipore, PEZGS0416) overnight in complete growth media, DMEM or RPMI respectively. 24 hr post-seeding, cells were starved in media containing 0.5% FBS + 1% Pen/Strep. 16 hr later, media was replaced with either DMEM/RPMI containing a total of 2 g/L glucose as follows: for U2Os cells low-glucose DMEM + 1 g/L D-(+)-Glucose + 10% FBS + 1% Pen/Strep was used; and for 66cl4 RPMI + 1 g/L D-(+)-Glucose + 10% FBS + 1% Pen/Strep. Cells were incubated for 12 hr at 37°C 5% CO2 and then prepared for immunofluorescence staining. (B) For MM6, cells were grown overnight at 1 million cells/ml in RPMI containing 10% FBS and 1% Pen/Strep. On the next day, cells were washed twice with 1x PBS (Wisent Biologicals, Cat# 311–010 CL) and resuspended in RPMI containing a total of 2 g/l glucose + 10% FBS + 1% Pen/Strep and incubated at 37°C 5% CO2 for 4 hr after which prepared for fluorescence staining. CD34+ cells were purchased from ATCC and primary high-eIF4E AML samples were obtained from our phase I clinical trial of ribavirin and low-dose cytarabine (*Assouline et al., 2009*; *2015*).

## Hyaluronidase treatment

One hour prior to the end of the 12 or 4 hr incubation of U20s, 66cl4 or MM6 cells, respectively, in media containing 2 g/lL glucose, cells where treated with Hyaluronidase from *Streptomyces Hyalurolyticus* (Sigma Aldrich, Cat# H1136) at 12 units/ml/1 million cell and incubated at 37°C 5% $CO_2$ for 1 hr. Hyaluronidase preparation: 1 Ampule (equivalent to 300 units) was resuspended at 1 unit/µl in the following buffer (20 mM sodium phosphate buffer, pH 7.0, with 77 mM sodium chloride and 0.1 mg/ml BSA at 1 mg/ml), incubated at room temperature for 10 min and then aliquoted and stored at −80°C. After

## Immunofluorescence, fluorescence and laser-scanning confocal microscopy

For HA staining, biotinylated hyaluronic acid binding protein (from Bovine Nasal Cartilage, Millipore, Cat# 385911) resuspended in 100 µl water: glycerol (50:50) was used. Fluorescence staining was carried out as described (PEG Protocol Cleveland Clinic). Briefly, following incubation with media containing 2 g/l glucose and treatment with Hyaluronidase, cells were washed thrice with complete growth media followed by three washes with 1 x PBS. Slides were then air-dried for 15 min and fixed with 4% Paraformaldehyde (32% solution from Electron Microscopy Sciences, PA, U.S.A, Cat# 15714) prepared in 1x PBS for 10 min at room temperature. After fixation, slides were air-dried and stained right away. Note that slides can be stored at 4°C for few days; however, longer term storing at 4°C or −20°C is not recommended as HA staining is lost due oxidative or mechanical breakdown of HA chains. For staining, slides were blocked for 1 hr at room temperature in blocking solution (10% FBS +0.2% Triton-X-100 in 1x PBS) and incubated with HABP (1:100 dilution in blocking solution) overnight at 4°C; followed by three washes in blocking solution (5 min each). Slides were then incubated with FITC or Texas Red conjugated Streptavidin (Vector Laboratories, CA, U.S.A, Cat# SA-1200) (1:500 dilution in blocking solution) for 1 hr at room temperature; washed four times with 1x PBS (pH 7.4) and mounted in Vectashield with DAPI (Vector laboratories, Cat# H-1200). For CD44 staining: mouse anti-CD44 (A3D8 or 156–3 C11) was used at 1:500 dilution in blocking buffer, respectively. Secondary anti-mouse FITC antibody (Jackson Laboratories, PA, U.S.A) was used at 1:500 dilution. Incubations with primary antibodies were carried out overnight at 4°C and incubations with secondary antibodies were done for 1 hr at room temperature.

Analysis was carried out using a laser-scanning confocal microscope (LSM700 META; Carl Zeiss, Inc.), exciting 405 and 543 nm or 488 nm with a 40x or 63x objective and 2x digital zoom (where indicated), and numerical aperture of 1.4. Channels were detected separately, with no cross talk observed. Confocal micrographs represent single sections through the plane of the cell. Images were obtained from ZEN software (Carl Zeiss, Inc.) and displayed using Adobe Photoshop CS6 (Adobe). For quantification of HA intensity, ZEN software was used to assess the staining intensity of HA and DAPI per cell in a given image. Fifty cells having similar DAPI intensities were chosen per condition to calculate the geometrical means and the standard deviation of HA intensity.

## Ribavirin treatment

Treatment of U2Os, MM6 and 66cl4 cells with ribavirin was carried out as follows. For U2Os and 66cl4 cell lines: cells were seeded at 0.7 million (Untreated) or 1.4 million cells (Ribavirin treated) per 10 cm plate. 20 µM ribavirin was used to treat cells for 48 or 96 hr (for 96 hr treatments, ribavirin is replenished every 24 or 48 hr). For MM6 cells were treated with 5 µM ribavirin at 0.5 million/ml (Untreated) or 1 million/ml (Ribavirin treated) cell density, and ribavirin was replenished every 24 hr for 48 or 96 hr.

## In vitro fluoroblok matrigel invasion assay

Fluoroblok Invasion assays were performed according to manufacturer instructions (*Partridge and Flaherty, 2009*). Briefly, fluoroblok 24-well inserts with 8-micron pore size PET membrane (Corning, MA, U.S.A, Cat # 351152) were precoated with 1 µg/µl Matrigel matrix basement membrane (BD Biosciences Cat # 356237) diluted in serum free DMEM media for 24 hr at 37°C. Cells were harvested, centrifuged, rinsed three times with serum free media and suspended at a density of $1 \times 10^5$ cells/ 300 µl in culture media containing 0.5% FBS. Cells were then plated on Matrigel coated and uncoated inserts and 750 µl of culture media containing 10% FBS was added to the lower compartment of the chamber. Chambers were incubated at 37°C for 48 hr. After invasion period, cells were labeled with DilC12(3) perchlorate, ultra-pure (Enzo Life Sciences, NY, U.S.A, Cat # ENZ-52206) diluted at 1:2000 in culture media for 10 min at 37°C followed by 15 min incubation at 4°C. After washing, fluorescence of invaded and migrated cells was measured at wavelengths of 549/565 nm (Ex/Em) on a bottom-reading fluorescent plate reader and images were taken using an inverted fluoresce microscope to verify results. Data are expressed according to the following equation:

$$\% \, invasion = \frac{\text{Mean RFU of cells invaded through matrigel coated inserts towards chemoattractant}}{\text{Mean RFU of cells migrated though uncoated inserts towards chemoattractant}}$$

## In vitro scratch assay

U2Os cells overexpressing 2Flag-eIF4E or vector control subjected to HAS3 knockdown for 72 hr were seeded in Millicell EZ slide (4-well glass) at 80,000 cell per well diluted in DMEM containing 10% FBS + 1% Pen/Step. On the next day, cells were starved in DMEM containing 0.5% FBS + 1% Pen/Strep for 16 hr. Following incubation, a scratch was made in the cell layer using a 1 ml tip. Cells were then washed with PBS to remove floating cells and fresh DMEM-low-glucose media containing 10% FBS + 1% Pen/Strep + 1 g/l glucose was added (total glucose concentration is 2 g/l). For the HAse treatment conditions, 12 units/ml of *Streptomyces* Hyaluronidase was added per well following media change. Pictures of the scratches were taken at the time of the scratching and after 16 hr. The area not filled with cells was quantified using TScratch software (available from Computational Science and Engineering lab at ETH University, Zurich, Switzerland).

## Fluorophore-assisted carbohydrate electrophoresis (FACE)

U2Os cells were seeded at 1 million cells/well of a six-well plate and incubated at 37°C 5% $CO_2$ overnight. 24 hr after, cells were starved in DMEM containing 0.5% FBS + 1% Pen/Strep for 16 hr. Media was then replaced with low glucose DMEM + 1 g/l glucose + 10% FBS + 1% Pen/Strep and incubated for 12 hr at 37°C, 5% CO2. Following incubation, samples were prepared for FACE analysis as previously described (PEG protocol Cleveland Clinic). Briefly, media was collected and cells were washed 3 times with 1x PBS. Media and cells were treated with 1x Proteinase K (from Tritirachium album, Sigma Aldrich, Cat# P2308) for 2 hr at 60°C followed by precipitation and treatment with Hylauronidase and Chondroitinase ABC. Samples were finally lyophilized, resuspended in AMAC solution and separated on acrylamide gels as described. Fluorescence detection of AMAC derivatives was achieved with GelDoc system. DNA was quantified using Quant-iT PicoGreen dsDNA assay kit (ThermoFisher, P11496).

## Histochemistry staining

Tumor blocks of mouse mammary tumors and lung metastases of published data (*Pettersson et al., 2015*) were analyzed for HA staining as a function of ribavirin treatment from 10 ribavirin and 10 control animals. A total of $3 \times 4$ µm serial step sections were prepared and stained for each mouse for HABP and hematoxylin. Batch analysis of 10 selected areas of $5.8 \times 10^{-2}$ mm ($2.4 \times 105$ pixels)

were run. Percent positive nuclei per section were determined by addition of areas until the average percent positive nuclei for one section did not change. One tumor section per animal was analyzed, and at least three areas of healthy tissue were taken into consideration for the percent positive nuclei result.

### In situ protein synthesis assay

We used Click-iT HPG Alexa Four 488 (Thermo Fisher Scientific, Cat# C10428) according to the manufacturer's instructions. Briefly, cells pretreated with 20 µM Ribavirin for 48 hr were seeded in Millicell EZ slide (4-well glass) at 80,000 cell per well diluted in DMEM containing 10% FBS + 1% Pen/Step. On the next day, cells were starved in DMEM containing 0.5% FBS + 1% Pen/Strep for 16 hr. Following starvation, media was replaced with fresh DMEM-low glucose containing 10% FBS + 1% Pen/Strep+1 g/l glucose was added (total glucose concentration is 2 g/l) and cells were incubated for 12 hr to allow synthesis of HA. After incubation, cells were incubated with HPG for 1 hr in methionine-free RPMI + 10% FBS + 1% P/S + 1 g/l glucose. Cells were then washed with PBS and fixed with 3.7% formaldehyde for 15 min at room temperature and permeabilized with 0.5% Triton X-100. Incorporation of HPG was detected using Click-iT Cell Reaction Buffer Kit according to the manufacturer's instructions. Confocal analysis was used to assess in situ protein synthesis.

## Acknowledgements

This project was supported by grants to KLBB from the NIH (80728 and 98571) and the Leukemia and Lymphoma Translational Research Program. KLBB holds a Canada Research Chair in Molecular Biology of the Nucleus. Special thanks to the PEG funding (the project described was supported by Award Number P01HL107147 from the National Heart, Lung and Blood Institute. The content is solely the responsibility of the authors and does not necessarily represent the official views of the National Heart, Lung and Blood Institute or the National Institutes of Health) as well as the CIHR operating grant MOP-115002.

## Additional information

### Funding

| Funder | Grant reference number | Author |
|---|---|---|
| National Institutes of Health | 80728 | Katherine LB Borden |
| Leukemia and Lymphoma Society | | Katherine LB Borden |
| National Institutes of Health | 98571 | Katherine LB Borden |

The funders had no role in study design, data collection and interpretation, or the decision to submit the work for publication.

### Author contributions

Hiba Ahmad Zahreddine, Conceptualization, Data curation, Formal analysis, Validation, Investigation, Methodology, Writing—original draft, Writing—review and editing, Designed and performed experiments, Analysed data; Biljana Culjkovic-Kraljacic, Conceptualization, Resources, Formal analysis, Supervision, Validation, Investigation, Visualization, Methodology, Project administration, Writing—review and editing, Designed and performed experiments, Analysed data; Audrey Emond, Resources, Methodology, Performed experiments, Revised manuscript; Filippa Pettersson, Resources, Methodology, Performed experiments; Ronald Midura, Resources, Supervision, Validation, Visualization, Methodology, Writing—review and editing, Designed experiments, Analysed data; Mark Lauer, Resources, Validation, Visualization, Methodology, Writing—review and editing, Performed experiments; Sonia Del Rincon, Resources, Methodology, Writing—review and editing, Performed experiments; Valbona Cali, Resources, Supervision, Methodology, Performed experiments; Sarit Assouline, Conceptualization, Resources, Funding acquisition, Investigation, Project administration, Edited manuscript; Wilson H Miller, Resources, Methodology, Writing—review and editing, Analyzed data;

Vincent Hascall, Resources, Supervision, Writing—review and editing, Designed experiments, Analysed data; Katherine LB Borden, Conceptualization, Resources, Formal analysis, Supervision, Funding acquisition, Investigation, Methodology, Writing—original draft, Project administration, Writing—review and editing, Designed experiments, Analysed data

### Author ORCIDs
Hiba Ahmad Zahreddine (iD) https://orcid.org/0000-0001-8905-1661
Katherine LB Borden (iD) http://orcid.org/0000-0003-2188-5074

### Ethics
Human subjects: Written informed consent was obtained in accordance with the Declaration of Helsinki. This study received IRB approval from the Conseil d'évaluation éthique pour les recherches en santé (CERES) (approval numbers 13-089-CERES and 14-112-CERES) and the Comité d'éthique de la faculté de Medicine (CERFM#195; tissue bank). The study was also approved by Health Canada (112878, 132348 and 173149; samples taken from three different protocols). ClinicalTrials.gov registry numbers: NCT00559091, NCT01056523 and NCT02073838.

### Decision letter and Author response
Decision letter https://doi.org/10.7554/eLife.29830.017
Author response https://doi.org/10.7554/eLife.29830.018

## Additional files

### Supplementary files
• Transparent reporting form
DOI: https://doi.org/10.7554/eLife.29830.014

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
