## [Decision Letter]

Thank you for submitting your article "The eukaryotic translation initiation factor eIF4E drives production of hyaluronan" for consideration by *eLife*. Your article has been favorably evaluated by Kevin Struhl (Senior Editor), Alan Hinnebusch (Reviewing Editor), and three reviewers. The reviewers have opted to remain anonymous.

The reviewers have discussed the reviews with one another and the Reviewing Editor has drafted this decision to help you prepare a revised submission.

Summary:

This paper deals with the mechanism of oncogenic transformation promoted by overexpression of the cap binding protein eIF4E, and evidence that post-transcriptional control by eIF4E of hyaluronic acid producing enzymes and receptors has an important role in the process. They used RNA-IP of nuclear extracts to identify mRNAs bound to eIF4E in the nucleus, which included multiple hyaluronic acid (HA)-synthesizing enzymes and HA receptors, and showed evidence that nuclear export of all but one of these mRNAs was enhanced by overexpressing WT but not an export-defective eIF4E mutant. For several of these mRNAs, they showed that increased protein expression accompanied increased mRNA export. This appeared to involve increased translation of the mRNAs in addition to increased mRNA export as protein expression was increased by a smaller amount when the export-defective eIF4E was overexpressed. Consistent with this, they showed that cellular HA levels were increased by overexpression (OE) of WT, and less so by the export-defective mutant 4E protein, and the HA was found coating the cell surface. siRNA KD of eIF4E, or its inhibition with ribavirin, eliminated the HA overproduction conferred by eIF4E OE. They extended these assays to various malignant cell lines and demonstrated eIF4E IP with HA-relevant mRNAs and eIF4E-dependent HA overproduction on cell surfaces, supporting the relevance to cancer of the results obtained from eIF4E OE in U2OS cells. They go on to show that increased invasiveness of cells conferred by OE of WT eIF4E, and less so by the mutant eIF4E is reversed by siRNA KD of the HA biosynthetic enzyme HAS3, which reduces HA expression, to same extent observed for KD of eIF4E or ribavirin treatment, indicating that the known increased invasiveness conferred by eIF4E OE is dependent on HA production. As KD of HAS3 reduces invasiveness even in cells not OE eIF4E, there is some a question about whether the effect of HAS3 KD merely reflects a requirement for HA in invasiveness, and whether other targets of eIF4E mediate the increased invasiveness of eIF4E OE. In fact, inhibiting eIF4E with ribavirin inhibits invasiveness even when cells are treated with HAse to digest HA, implying additional eIF4E targets beyond the enzymes of HA biosynthesis. They further showed that increased cell migration in a wound-healing assay conferred by eIF4E overexpression was blocked by KD of HAS3 or HAase treatment to reverse HA production. Previously, they showed that ribavirin reduced metastasis in an eIF4E-dependent mouse lung tumor cancer model, and they showed here that ribavirin reduced HA levels in the lungs of these mice. Finally, they showed that cell surface receptor CD44, a known downstream effector of HA, is overproduced in cells overexpressing eIF4E, as would be expected from their RNA-IP and mRNA export assays, and that siRNA KD of CD44 reduced the increased invasiveness conferred by eIF4E OE, without affecting the status of HA itself. They conclude that the HA network (biosynthetic enzymes and downstream effectors) is regulated post-transcriptionally by eIF4E, at least in normal or malignant cells that are overexpressing eIF4E. These results are significant in providing evidence for specific target genes of eIF4E required for the known oncogenic effects of overexpressing this initiation factor.

Essential revisions:

1) The results of the initial RIP screen described in the beginning of the Results should be presented as supplementary information.

2) The complete Western data in Figure 1—figure supplement 1, containing the MCl-1 loading control, should be incorporated into Figure 1.

3) The Western blots of eIF4E recovered in immune complexes with eIF4E antibodies and IgG relative to input amount of eIF4E should be shown as necessary documentation of the RIPs in Figure 1, as well as for those in Figure 3. Also, to more accurately evaluate the level of mRNA enrichment, the percent IP compared to input could be shown for the RNAs immunoprecipitated in Figure 1.

In relation to Figure 3, showing enrichment of HA pathway mRNAs associated with eIF4E in AML-M5 cells that overexpress eIF4E, control pulldowns from the BM CD34+ cells are required to show that these messages are not as enriched in this low-4E control cell line; and 4E blots from these two lines should also be provided.

As there is no quantification of metastases in Figure 5, either previous measurements should be more thoroughly cited or such quantifications should be conducted here.

In relation to results in Figure 2, the authors could treat with the eIF4E/eIF4G interaction inhibitor 4EGI-1 to show that the residual expression is reduced by inhibiting the eIF4E-eIF4G interaction, to provide evidence that 4E overexpression leads to up-regulation of translation of HA-pathway proteins. Alternatively, and more incisive, the authors could attempt to provide evidence that eIF4E overexpression increases the rate of translation initiation of HA-pathway mRNAs by conferring shift towards heavier polysomes of the polysome-associated pool of these mRNAs on eIF4E overexpression. Although it is not necessary to justify the main conclusions of the paper, providing evidence for altered translation efficiencies of HA pathway proteins on 4E overexpression, or lack thereof, would considerably enhance the scientific quality of the paper.

*Reviewer #1:*

This paper deals with the mechanism of oncogenic transformation promoted by overexpression of the cap binding protein eIF4E, and evidence that post-transcriptional control by eIF4E of hyaluronic acid producing enzymes and receptors has an important role in the process. They used RNA-IP of nuclear extracts to identify mRNAs bound to eIF4E in the nucleus, which included multiple hyaluronic acid (HA)-synthesizing enzymes and HA receptors, and showed evidence that nuclear export of all but one of these mRNAs was enhanced by overexpressing WT but not an export-defective eIF4E mutant. For several of these mRNAs, they showed that increased protein expression accompanied increased mRNA export. This appeared to involve increase translation of the mRNAs in addition to increased mRNA export as protein expression was increased by a smaller amount when the export-defective eIF4E was overexpressed. Consistent with this, they showed that cellular HA levels were increased by overexpression (OE) of WT, and less so by the export-defective mutant 4E protein, and the HA was found coating the cell surface. siRNA KD of eIF4E, or its inhibition with ribavirin, eliminated the HA overproduction conferred by eIF4E OE. They extended these assays to various malignant cell lines and demonstrated eIF4E IP with HA-relevant mRNAs and eIF4E-dependent HA overproduction on cell surfaces, supporting the relevance to cancer of the results obtained from eIF4E OE in U2OS cells. They go on to show that increased invasiveness of cells conferred by OE of WT eIF4E, and less so by the mutant eIF4E is reversed by siRNA KD of the HA biosynthetic enzyme HAS3, which reduces HA expression, to same extent observed for KD of eIF4E or ribavirin treatment, indicating that the known increased invasiveness conferred by eIF4E OE is dependent on HA production. As KD of HAS3 reduces invasiveness even in cells not OE eIF4E, there is some a question about whether the effect of HAS3 KD merely reflects a requirement for HA in invasiveness, and whether other targets of eIF4E mediate the increased invasiveness of eIF4E OE. In fact, inhibiting eIF4E with ribavirin inhibits invasiveness even when cells are treated with HAse to digest HA, implying additional eIF4E targets beyond the enzymes of HA biosynthesis. They further showed that increased cell migration in a wound-healing assay conferred by eIF4E overexpression was blocked by KD of HAS3 or HAase treatment to reverse HA production. Previously, they showed that ribavirin reduced metastasis in an eIF4E-dependent mouse lung tumor cancer model, and they showed here that ribavirin reduced HA levels in the lungs of these mice. Finally, they showed that cell surface receptor CD44, a known downstream effector of HA, is overproduced in cells overexpressing eIF4E, as would be expected from their RNA-IP and mRNA export assays, and that siRNA KD of CD44 reduced the increased invasiveness conferred by eIF4E OE, without affecting the status of HA itself. They conclude that the HA network (biosynthetic enzymes and downstream effectors) is regulated post-transcriptionally by eIF4E, at least in normal or malignant cells that are overexpressing eIF4E.

These results seem significant in identifying specific targets of eIF4E required for the known oncogenic effects of overexpressing eIF4E, which typically has been attributed to a heightened requirement of oncogenic mRNAs for eIF4E and its associated factors in the eIF4F complex, particularly the helicase eIF4A, for translation initiation, owing to the presence of RNA structures in these mRNAs that impair ribosome attachment or scanning. The ability of eIF4E OE to enhance HA production seems to involve both increased export and increased translation of the mRNAs encoding HA network components, as OE of the transport-defective eIF4E variant still conferred increased protein production. However, there is no direct evidence for increased translation of any of these mRNAs, and thus, no examination of what sequences or structures might confer increased translation on eIF4E OE. Nor is there any molecular understanding of what sequences/structures in these mRNAs confer increased nuclear export at high eIF4E levels. Nevertheless, the findings described in the Abstract, which appear to be well established by careful experimentation, do shed new light on the mechanism of oncogenesis in response to eIF4E overexpression.

1) It would enhance the scientific quality of the paper if they provided evidence that some of the mRNAs in the HA network whose nuclear export is enhanced by eIF4E overexpression were shown to exhibit increased rates of translation initiation as indicated by a shift towards heavier polysomes of the polysome-associated pool of these mRNAs on eIF4E overexpression. (Increased export alone would be expected to increase the fraction of mRNA engaged with ribosomes but not increase the number of ribosomes per mRNA.) If so, a consideration of the potential for forming secondary structure of the mRNA 5'UTRs would be in order to determine whether these mRNAs appear likely to have a greater than average dependence on eIF4F/eIF4A for removing RNA structures that impede ribosome attachment or scanning.

*Reviewer #2:*

The manuscript "The eukaryotic translation initiation factor eIF4E drives production of hyaluronan" provides data showing that eIF4E promotes the increased protein production of hyaluroan and associated proteins through mRNA export and post-transcriptional control. This work is important with regards to understanding the role of elevated eIF4E in cancer and should be published if the reviewers meet the following concerns.

1) The authors introduce the fact that preliminary studies using genome-wide screens of nuclear eIF4E RIPs provided the evidence that HA biosynthesis factor RNAs were enriched with eIF4E however they do not show any of the data in this paper. This is strange as it is the prerequisite for the rest of the paper. Although the RT-qPCR in Figure 1 shows that these RNAs are enriched, the preliminary screen should also be shown. In addition, do the authors have any data to suggest that these RNA targets are direct, e.g. cross-linking data?

2) In Figure 1 legend the authors state that MCl-1 served as a loading control, however the western blots do not contain an MCl-1 western. This needs to be added. Further to this, it is unclear why the more extensive panel of western blots was not performed/included in the main Figure 1 and instead used in the supplemental figure. This would improve the impact of the figure by showing the effects on translation on more of these transcripts depicted in Figure 1.

3) In Figure 2 the authors show that HA is decreased in the 2FS53A expressing cells. It would be a good addition to treat with the eIF4E/eIF4G interaction inhibitor 4EGI-1 to show that the residual expression is reduced when you inhibit eIF4E-eIF4G interaction. This should also be performed on the 2F4E background to show that HA expression is particularly sensitive to perturbing this interaction.

4) In Figure 3 the authors look at the enrichment of HA pathway mRNAs on nuclear eIF4E RIP and show that in the AML-M5 cells that overexpress eIF4E, these RNAs are enriched. Do the authors have complementary control pulldowns from the BM CD34+ cells to show that these messages are not as enriched in this low 4E control cell line? It would also be advantageous to show 4E blots from these two lines.

5) The authors state that 'eIF4E targeting leads to reduced HA levels and decreased metastases in vivo'. There is no quantification of metastases in Figure 5 and so although the HA statement holds true from data in Figure 5, the second part of this statement pertaining to metastases has not been presented within the figure.

*Reviewer #3:*

In this study by Zahreddine et el., the authors have shown that eukaryotic translation initiation factor eIF4E controls production of glycosaminoglycan Hyaluronan (HA), a major component of the tumor microenvironment and its downstream effectors post-transcriptionally. EIF4E was previously shown to modulate expression of many transcripts through its dual role in nuclear mRNA export and protein translation, both functions are required for its oncogenic role. The authors further showed that enzymes encoding the HA biosynthetic pathway, HAS3, CD44 and associated factors are eIF4E targets and demonstrated that this pathway is required by eIF4E to drive its oncogenic activities.

The results of this study are important for both cancer and gene expression fields, for understanding the impact and role of oncogenic eIF4E in cancer, as well as for characterizing targets of eIF4E driven export and translation. This manuscript demonstrated that the HA network is subjected to coordinated post-transcriptional control by eIF4E, and that eIF4E drives the production of cell-associated HA, which in turn, changes the cellular architecture, enabling the processes of tumor invasion and metastasis. This is an important manuscript with novel, significant findings and is a suitable candidate for publication in *eLife*. The authors need to address the following before the manuscript is accepted for publication.

1) The authors performed RNA immunoprecipitation (RIP) to identify eIF4E associated transcripts using ani-eIF4E antibodies and compared the results to IgG control (Figure 1). The authors need to clarify the text as to how the HA biosynthesis enzyme targets were identified and selected for validation by RT-PCR for Figure 1. It is unclear as a genome-wide screen is mentioned in the Results section (suggesting a microarray or seq after the RIP) or if these targets were selected based on the literature and then the selected HA pathway panel was screened by RIP and then RT-PCR.

2) The Western blot for the IP of eIF4E and IgG relative to input should be shown as a control for the RIP in Figure 1 as well as the western blot for the IP in Figure 3. To show the level of mRNA enrichment, percent IP compared to Input can be shown for RNAs immunoprecipitated in Figure 1.

3) As a control, given that different targets like GPI are not affected at the export level but at the translation level, while the other targets are affected at both levels, the authors could show the nuclear enrichment of the HA pathway targets that are affected at the export level upon RNAi or Ribavirin treatments.

4) Given that HA is expressed in normal cells as well, and ribavirin is used to block HA biosynthesis in malignant cells, the authors should provide comments on whether it is known that eIF4E controls HA pathway enzymes and CD44 translation in normal cells also, or whether these are not affected by ribavirin.

5) Based on their interesting FUNCAT results showing translation in the protrusions, the authors postulate that eIF4E could be involved in localized translation to spatially couple translation of HA pathway enzymes with HA biosynthesis. The authors should comment if it is known or show if eIF4E is localized in the protrusions, which would support this.

[Editors' note: further revisions were requested prior to acceptance, as described below.]

Thank you for resubmitting your work entitled "The eukaryotic translation initiation factor eIF4E harnesses hyaluronan production to drive its malignant activity" for further consideration at *eLife*. Your revised article has been favorably evaluated by Kevin Struhl (Senior Editor), and the Reviewing Editor, Alan Hinnebusch.

The manuscript has been improved but there are some remaining issues that need to be addressed before acceptance, as outlined below:

1) In responding to the request that the results of the initial RIP screen be presented as supplementary information, the authors have eliminated all mention of the two screens that suggested that the enzymes of HA synthesis represent a coherent group of functionally related mRNAs dependent on eIF4E for export in cells overexpressing eIF4E, because the screens were only done once and rely on microarray versus RNA-seq technology. This is not an ideal solution as the experiments done to validate the HA mRNAs as eIF4E export targets now come out of the blue. It would be preferable to revert to the strategy of the original manuscript, describing the screens and the preliminary results they provided, but not in the Materials and methods section that the complete lists of hits from these screens is not being provided because the screens were done only once and thus lack statistical power. (The use of microarrays in itself does not disqualify the screens, as microarray analysis is actual more powerful than RNA-seq for mRNAs expressed at low levels.)

2) In revised Figure 1, the identity of the lower bands in the eIF4E Western blots need to be indicated in the legend.

3) Regarding the sentence: "We did not observe 141 substantial shifts for most of these mRNAs on the polysomes with modest shifts to heavier 142 polysomes including UAP, UGDH and UGP2 (Figure 1—figure supplement 3).", it appears that CD44 also shows evidence of increased polysome association.

---

## [Author Response]

Essential revisions:1) The results of the initial RIP screen described in the beginning of the Results should be presented as supplementary information.

We referenced our preliminary set of screens we used to identify candidate eIF4E mRNA export targets. The first screen was an RNA immunoprecipitation (RIP) from the nuclear fractions of U2Os cells using an anti-eIF4E antibody and identifying associated RNAs using microarray (RIP-Chip). The second screen determined which RNAs accumulated in the nucleus upon inhibition of eIF4E-dependent mRNA export with ribavirin using microarrays in THP1 AML cells, which are characterized by high-eF4E. In this screen, ribavirin treated or untreated cells were fractionated into nuclear and cytoplasmic components. Total mRNAs were also assessed and did not change as a function of ribavirin treatment. Data were presented as the ratio of RNA in the nuclear versus cytoplasmic fraction as a function of ribavirin treatment. Thus, higher fold enrichment indicates more RNA is accumulated in the nucleus upon ribavirin treatment due to inhibition of eIF4E-dependent mRNA export. Transcripts that are both enriched in the nuclear eIF4E immunoprecipitation (RIP) and have their nuclear export inhibited by ribavirin (as observed by nuclear accumulation of the transcript) relative to untreated controls were scored as positive hits and prioritized for independent validation by RT-qPCR as presented in Figure 1 of the manuscript. Strikingly, our analysis showed that nearly all the enzymes involved in hyaluronic acid (HA) synthesis were positive hits (shown in Author response image 1). For comparison, several negative controls are shown: tubulin, VCAM, GAPDH and UGT2B7.

These RIP-chip studies served as a launch point for the studies reported here. The purpose of these screens was to identify targets that we would validate independently using RT-qPCR, as we did in this manuscript. However, both screens were only done once and thus lack statistical power and relied on microarray technology, which is outdated with many disadvantages relative to RNA-Seq etc.

Our RT-qPCR studies reported in our manuscript validated our choice of studying HA in both U2Os and AML cells (shown in Figure 1 and Figure 1—figure supplement 1, Figure 3 and Figure 3—figure supplement 1). However, the preliminary nature of the screen studies makes us uncomfortable publishing the full list. Thus, we included a figure for the reviewers for the relevant factors in the pathway, but respectively request that these data not be included. We are currently working on similar experiments by RNA-Seq (in three biological replicates) but believe this to be beyond the scope of the current manuscript.

To avoid possible confusion over this, we have removed reference to these preliminary screens in the manuscript.

2) The complete Western data in Figure 1—figure supplement 1, containing the MCl^-^1 loading control, should be incorporated into Figure 1.

The Western blot data in Figure 1—figure supplement 1 has been incorporated into Figure 1. We note that Mcl1 is an established mRNA export target of eIF4E and as such serves as a positive control for eIF4E overexpression (Volpon, L. et al. Proc Natl Acad Sci U S A., 10;113(26):5263-8 (2016)).

3) The Western blots of eIF4E recovered in immune complexes with eIF4E antibodies and IgG relative to input amount of eIF4E should be shown as necessary documentation of the RIPs in Figure 1, as well as for those in Figure 3. Also, to more accurately evaluate the level of mRNA enrichment, the percent IP compared to input could be shown for the RNAs immunoprecipitated in Figure 1.

Western blots for eIF4E IPs are now shown in Figure 1 and Figure 3, as requested.

In parallel to the RIPs versus IgG (Figure 1), RIPs compared to input for RNAs are now presented in Figure 1—figure supplement 1. Comparison of fold changes for the given normalization methods shows that while normalizing to input controls produces higher fold changes for some transcripts, enrichment of others in the immunoprecipitated fraction is underestimated. Normalizing to input can lead to false negatives especially for very abundant RNAs. For instance, if every eIF4E is bound by a certain RNA, but that RNA is very abundant, one might falsely assume that it is not a target because it has a small depletion versus input. As such, normalizing levels of mRNA enrichment to input may not be more accurate when compared to IgG controls. IgG controls also provide important information about background binding to the beads, which is not incorporated when input alone is used. A discussion is added to the text regarding this data

In relation to Figure 3, showing enrichment of HA pathway mRNAs associated with eIF4E in AML-M5 cells that overexpress eIF4E, control pulldowns from the BM CD34+ cells are required to show that these messages are not as enriched in this low-4E control cell line; and 4E blots from these two lines should also be provided.

We note that CD34+ cells are primary cells from healthy human donors and not a cell line. We require a minimum of 50 million cells to generate a at least 0.5 mg of nuclear protein lysate per IP. Normal human bone marrow derived CD34+ cells are purchased commercially (Lonza or ATCC are the only suppliers in Canada) and the cost of doing 3 healthy donors (for cells alone) for this proposed experiment is over $ 100,000 USD. Unfortunately, there is no “normal” CD34+ cell line available. The AML cell line referred to by the referee used for the IP was MM6 AML cell lines, and thus amounts of material were not an issue. The primary CD34+ and primary AML studies in the manuscript explored HA levels using confocal microscopy, requiring few cells and thus not very expensive compared to the IP. While we appreciate that the proposed IP with CD34+ cells is an interesting experiment to do, unfortunately, it is not financially feasible for us to do it.

As there is no quantification of metastases in Figure 5, either previous measurements should be more thoroughly cited or such quantifications should be conducted here.

The original Cancer Research paper carried out detailed quantification for the metastatic burden for these animals. We now more completely describe this analysis in the text.

In relation to results in Figure 2, the authors could treat with the eIF4E/eIF4G interaction inhibitor 4EGI-1 to show that the residual expression is reduced by inhibiting the eIF4E-eIF4G interaction, to provide evidence that 4E overexpression leads to up-regulation of translation of HA-pathway proteins. Alternatively, and more incisive, the authors could attempt to provide evidence that eIF4E overexpression increases the rate of translation initiation of HA-pathway mRNAs by conferring shift towards heavier polysomes of the polysome-associated pool of these mRNAs on eIF4E overexpression. Although it is not necessary to justify the main conclusions of the paper, providing evidence for altered translation efficiencies of HA pathway proteins on 4E overexpression, or lack thereof, would considerably enhance the scientific quality of the paper.

We note that many nuclear proteins bind to the dorsal surface of eIF4E including LRPPRC one of the key mediators of eIF4E dependent mRNA export (see Volpon et al., RNA 2017). The LRPPRC-binding site on eIF4E overlaps with the eIF4G binding site suggesting that there is every possibility that 4EGI-1 could inhibit mRNA export functions of eIF4E as well as its functions in translation. Given this could confound our data interpretation, we opted for the polysome analysis. We compared vector control and eIF4E-overexpressing U2Os cells.

We observed shifts to heavier polysomes for positive control c-Myc RNAs and no apparent shifts for negative control Actin RNA upon eIF4E overexpression (similar to results in Rousseau et al., 1996, PNAS). Some of the HA pathway enzymes were modestly affected; while most were not altered substantially. For instance, eIF4E shifts UGP2 and UAP RNAs to heavier polysomes but we note these shifts are not dramatic. Our findings are consistent with studies with the S53A mutant, which is active in translation and not in export. By western blot, S53A produced mild stimulations to these same proteins, but given the more dramatic effects of wildtype eIF4E, the export was still the more substantial contributor to the observed phenotypes in U2Os cells. These data are shown in Figure 1—figure supplement 3.

Reviewer #1:[…] 1) It would enhance the scientific quality of the paper if they provided evidence that some of the mRNAs in the HA network whose nuclear export is enhanced by eIF4E overexpression were shown to exhibit increased rates of translation initiation as indicated by a shift towards heavier polysomes of the polysome-associated pool of these mRNAs on eIF4E overexpression. (Increased export alone would be expected to increase the fraction of mRNA engaged with ribosomes but not increase the number of ribosomes per mRNA.) If so, a consideration of the potential for forming secondary structure of the mRNA 5'UTRs would be in order to determine whether these mRNAs appear likely to have a greater than average dependence on eIF4F/eIF4A for removing RNA structures that impede ribosome attachment or scanning.

See Essential revisions section for addition of polysomes.

No obvious structural motifs were apparent when studying the 5’ UTRs of these transcripts. Most were longer than 150 nucleotides and according to m-fold would have the propensity for secondary structure. However, a complication is that many of the transcripts have multiple isoforms, and thus we would need to define which isoforms were expressed in our system to better understand the structural features that would be relevant. We note that none of these were dramatically shifted on polysomes.

Reviewer #2:[…] 1) The authors introduce the fact that preliminary studies using genome-wide screens of nuclear eIF4E RIPs provided the evidence that HA biosynthesis factor RNAs were enriched with eIF4E however they do not show any of the data in this paper. This is strange as it is the prerequisite for the rest of the paper. Although the RT-qPCR in Figure 1 shows that these RNAs are enriched, the preliminary screen should also be shown. In addition, do the authors have any data to suggest that these RNA targets are direct, e.g. cross-linking data?

This has been addressed under Essential revisionssection above.

We have carried out one RIP with cross-linking (Author response image 2) and find that indeed, these are direct targets. This was only carried out one time and thus we show it to the reviewer as preliminary data.

**Author response image 2. respfig2:** 

2) In Figure 1 legend the authors state that MCl-1 served as a loading control, however the western blots do not contain an MCl-1 western. This needs to be added. Further to this, it is unclear why the more extensive panel of western blots was not performed/included in the main Figure 1 and instead used in the supplemental figure. This would improve the impact of the figure by showing the effects on translation on more of these transcripts depicted in Figure 1.

MCl-1 is a positive control, we have clarified that text in this regard. It serves as an established mRNA export target of eIF4E (Volpon, L. et al. Proc Natl Acad Sci U S A., 10;113(26):5263-8 (2016)). The blot has been moved as requested.

3) In Figure 2 the authors show that HA is decreased in the 2FS53A expressing cells. It would be a good addition to treat with the eIF4E/eIF4G interaction inhibitor 4EGI-1 to show that the residual expression is reduced when you inhibit eIF4E-eIF4G interaction. This should also be performed on the 2F4E background to show that HA expression is particularly sensitive to perturbing this interaction.

We used polysomes as described above to answer this question. Note also discussion above for 4EGI-1.

4) In Figure 3 the authors look at the enrichment of HA pathway mRNAs on nuclear eIF4E RIP and show that in the AML-M5 cells that overexpress eIF4E, these RNAs are enriched. Do the authors have complementary control pulldowns from the BM CD34+ cells to show that these messages are not as enriched in this low 4E control cell line? It would also be advantageous to show 4E blots from these two lines.

This has been addressed in the Essential revisionssection above.

5) The authors state that 'eIF4E targeting leads to reduced HA levels and decreased metastases in vivo'. There is no quantification of metastases in Figure 5 and so although the HA statement holds true from data in Figure 5, the second part of this statement pertaining to metastases has not been presented within the figure.

This has been addressed in the Essential revisionssection above.

Reviewer #3:[…] 1) The authors performed RNA immunoprecipitation (RIP) to identify eIF4E associated transcripts using ani-eIF4E antibodies and compared the results to IgG control (Figure 1). The authors need to clarify the text as to how the HA biosynthesis enzyme targets were identified and selected for validation by RT-PCR for Figure 1. It is unclear as a genome-wide screen is mentioned in the Results section (suggesting a microarray or seq after the RIP) or if these targets were selected based on the literature and then the selected HA pathway panel was screened by RIP and then RT-PCR.

This has been addressed in the Essential revisionssection above.

2) The Western blot for the IP of eIF4E and IgG relative to input should be shown as a control for the RIP in Figure 1 as well as the western blot for the IP in Figure 3. To show the level of mRNA enrichment, percent IP compared to Input can be shown for RNAs immunoprecipitated in Figure 1.

The correction has been made, see Essential revisions section above.

3) As a control, given that different targets like GPI are not affected at the export level but at the translation level, while the other targets are affected at both levels, the authors could show the nuclear enrichment of the HA pathway targets that are affected at the export level upon RNAi or Ribavirin treatments.

We showed that in AML cell line MM6, ribavirin treatment reduced nuclear eIF4E immunoprecipitation and reduced export of HA pathway targets, but not negative controls. This led to reduced HA as expected. This data is in Figure 3 and Figure 3—figure supplement 1.

4) Given that HA is expressed in normal cells as well, and ribavirin is used to block HA biosynthesis in malignant cells, the authors should provide comments on whether it is known that eIF4E controls HA pathway enzymes and CD44 translation in normal cells also, or whether these are not affected by ribavirin.

We now include this discussion in our conclusions. Our data suggest that HA levels would scale with both eIF4E levels and loss of eIF4E control (which also contributes to malignancy). Thus in normal cells with less eIF4E and tighter eIF4E regulation, HA biosynthesis would be expected to be lower. In AML cells, where eIF4E levels are higher and regulatory mechanisms are impaired such as by the loss of the nuclear regulator PRH/Hex (Topisirovic et al., EMBO 2003), larger fractions of these RNAs can become eIF4E bound in the nucleus and be exported. It would be expected that ribavirin treatment would reduce HA biosynthesis in normal cells, but unfortunately we can barely detect HA in normal cells and thus would have to improve the sensitivity of our confocal assay in order to directly test this.

5) Based on their interesting FUNCAT results showing translation in the protrusions, the authors postulate that eIF4E could be involved in localized translation to spatially couple translation of HA pathway enzymes with HA biosynthesis. The authors should comment if it is known or show if eIF4E is localized in the protrusions, which would support this.

Our data from fluorescence immunostaining with anti-eIF4E antibody indicate that eIF4E could be localized in HA rich protrusions. However, extension of this result to understand the role of eIF4E here by co-staining with FUNCAT and eIF4E has proven to be technically challenging and requires further optimization with particular regard to fixation conditions.

[Editors' note: further revisions were requested prior to acceptance, as described below.]

1) In responding to the request that the results of the initial RIP screen be presented as supplementary information, the authors have eliminated all mention of the two screens that suggested that the enzymes of HA synthesis represent a coherent group of functionally related mRNAs dependent on eIF4E for export in cells overexpressing eIF4E, because the screens were only done once and rely on microarray versus RNA-seq technology. This is not an ideal solution as the experiments done to validate the HA mRNAs as eIF4E export targets now come out of the blue. It would be preferable to revert to the strategy of the original manuscript, describing the screens and the preliminary results they provided, but not in the Materials and methods section that the complete lists of hits from these screens is not being provided because the screens were done only once and thus lack statistical power. (The use of microarrays in itself does not disqualify the screens, as microarray analysis is actual more powerful than RNA-seq for mRNAs expressed at low levels.)2) In revised Figure 1, the identity of the lower bands in the eIF4E Western blots need to be indicated in the legend.3) Regarding the sentence: "We did not observe 141 substantial shifts for most of these mRNAs on the polysomes with modest shifts to heavier 142 polysomes including UAP, UGDH and UGP2 (Figure 1—figure supplement 3).", it appears that CD44 also shows evidence of increased polysome association.

We incorporated all the changes as recommended.